# Inversion tectonics: a brief petroleum industry perspective

Gábor Tari[1], Didier Arbouille[2], Zsolt Schléder[1], Tamás Tóth[3]

[1]OMV Upstream, Exploration, 1020 Vienna, Austria
[2]IHS Markit, Geneva, Switzerland
[3]GeoMega Ltd, Budapest, Hungary

*Correspondence to*: Gabor Tari (gabor.tari@omv.com)

**Abstract.** Inverted structures provide traps for petroleum exploration, typically four-way structural closures. As to the degree of inversion, based on large number of worldwide examples seen in various basins, the most preferred petroleum
exploration targets are mild to moderate inversion structures, defined by the location of the null-points. In these instances, the closures have a relatively small vertical amplitude, but simple in a map-view sense and well imaged on seismic reflection data. Also, the closures typically cluster above the extensional depocentres which tend to contain source rocks providing petroleum charge during and after the inversion. Cases for strong or total inversion are generally not that common and typically are not considered as ideal exploration prospects, mostly due to breaching and seismic imaging challenges
associated with the trap(s) formed early on in the process of inversion. Also, migration may become tortuous due to the structural complexity or the source rock units may be uplifted above the hydrocarbon generation window effectively terminating the charge once the inversion occurred.

Cases of inversion tectonics can be grouped into two main modes. A structure developed in Mode I inversion if the syn-rift succession in the pre-existing extensional basin unit is thicker than its post-rift cover including the pre- and syn-inversion
part of it. In contrast, a structure evolved in Mode II inversion if the opposite syn- versus post-rift sequence thickness ratio can be observed. These two modes have different impacts on the petroleum system elements in any given inversion structure. Mode I inversion tends to develop in failed, intra-continental rifts and proximal passive margins and Mode II structures are associated with back-arc basins and distal parts of passive margins.

For any particular structure the evidence for inversion is typically provided by subsurface data sets such as reflection seismic
and well data. However, in many cases the deeper segments of the structure are either poorly imaged by the seismic data and/or have not been penetrated by exploration wells. In these cases the interpretation in terms of inversion has to rely on the regional understanding of the basin evolution with evidence for an early phase of crustal extension by normal faulting.

## 1 Introduction

Whereas the concept of structural inversion has been around for a century (e.g. Lamplugh, 1919), the term has been
specifically used for the first time by Glennie and Boegner (1981) to explain the evolution of the Sole Pit structure located in

the UK sector of the southern North Sea. At the same time, inversion structures callled "Sunda-type folds" were described by Eubank and Makki (1981) in Indonesia. The first generalized description of structural inversion was offered by Bally (1984) using a 3-step cartoon depicting an extensional half-graben subjected to subsequent contraction. Both the concept and the term of inversion tectonics gained rapid acceptance by the petroleum industry and the academia as shown by the large

number of papers produced on this subject in the 1980s and 1990s. In the two "classic" volumes on inversion tectonics by Cooper et al. (1989) and Buchanan and Buchanan, 1995) numerous case studies were published using data sets provided by petroleum exploration companies (e.g. Roberts, 1989; Hayward and Graham, 1989; Badley et al., 1989; Cartwright, 1989). In addition, detailed outcrop studies combined with a good understanding of the structural geology context, done almost exclusively in fold and thrust belts (e.g. Butler, 1989; de Graciansky et al., 1989; McClay et al., 1989; Cooper et al., 1995;

Flinch and Casas, 1996), offered an additional tool for recognizing inversion early on.

During the last 30 years many facets of inversion tectonics were addressed, including physical modelling (e.g. McClay, 1989, 1995; Mitra and Islam, 1994; Eisenstadt and Withjack, 1995; Keller and McClay, 1995; Yamada and McClay, 2004; Panien et al 2005; Amilibia et al., 2005; Bonini et al., 2011; Granado et al., 2017; Roma et al 2018; Ferrer et al., 2019), numerical modelling (e.g. Panien et al 2006; Buiter et al., 2007; Granado and Ruh, 2019 ), basin modelling (e.g. Neumaier et

al., 2016, 2019 ; Omodeo-Salé et al., 2019) and crustal-scale geodynamics ( e.g. Ziegler, 1987; Ziegler et al., 1989; Cloetingh, 2008). Obviously, as 3D seismic reflection data became frequently used by the petroleum industry, more subsurface case studies addressed inversion tectonics quantitatively (e.g. Davies et al., 2004; Jackson and Larsen, 2008; Jackson et al., 2013; Reilly et al., 2017; Phillips et al., 2020).

There is a paper devoted to salt tectonics and inversion in this Special Issue (Dooley and Hudec, 2020) and, therefore, the

very important role of salt tectonics in inversion tectonics will not be discussed in any details here, even though the cartoon by Bally (Fig. 1f) hints at the involvement of salt. Some of the key effects of salt during inversion include the decoupling of deformation above and below (e.g., Letouzey et al., 1995; Withjack and Callaway, 2000) and the sealing capacity of pre-, syn- and post-rift salt units impacting the hydrocarbon migration depending on the geodynamic setting on the salt basin (e.g. Rowan, 2014).

In this paper we provide a brief overview of inversion tectonics, specifically from the view point of the petroleum industry. The last 30 years saw lots of work done on the practical application of this important structural geology concept in the hydrocarbon exploration process. Whereas the impact of structural inversion became increasingly evident in many case studies, in our opinion, there is room for improvement in two major aspects. Firstly, the term of structural inversion is being used in a very broad sense across the industry which underlines the need to revisit the original definition of this process.

There has to be a clear distinction between regional-scale and individual structure (prospect)-specific inversion as these processes manifest themselves differently. Secondly, we observe an interesting disparity in the usage of structural inversion in the petroleum industry. During the life-cycle of many exploration and production projects the interpretation of the trap(s) in terms of structural inversion is preferentially used during the exploration phase as it has, in general, a positive connotation for prospectivity (see later). In contrast, during the appraisal and production phase the interpretation of the traps in a field in

terms of inversion, as a trap forming mechanism, typically becomes un(der)appreciated. We found that the description of existing fields generally lacks the reference to inversion tectonics as the trap forming mechanism, but instead, the trap itself is referred to as the result of reverse faulting or compression.

We provide below several case studies of regional-scale and prospect/field-scale structural inversion to illustrate the typical challenges of applying this important concept in the petroleum industry. We chose drilled prospects and hydrocarbon fields

from three very different basins to illustrate various aspects of the interpretational process of invoking inversion for the traps in these fields. The Budafa and Lovászi oil/gas fields in the Pannonian Basin of Hungary are onshore fields which were discovered in 1940 and are already depleted (Dank, 1985). There are numerous inversion structures in the Eastern Mediterranean, we show two of these located in the Egyptian offshore, the Mango and the Goliath anticlines. Another inversion structure in the same region, the giant offshore Tamar gas field in the Israeli sector of the Levant Basin was

discovered in 2009 and it started to produce just recently (Needham et al., 2017). The Atlantic margin of central Morocco also has lots of inverted structures due to the Cenozoic orogeny of the Atlas being superimposed on Mesozoic Tethyan rift systems.

Whereas there are major differences between these inversion structures, they are similar in the sense that their inverted nature may not be always determined by analysing them in isolation, but only in a regional geologic context. Therefore we

will zoom out from the areas of these individual structures and will highlight the regional aspects of the inversion in the respective basins where they are located.

We intentionally chose examples where the inversion anticlines have an equal or thicker post-rift sequence than the underlying syn-rift basin fill. As a new observational model, we suggest that the inversion tectonics should be categorized into two main modes. The seemingly more common, but certainly more widely recognized "classic" Mode I inversion is

where the syn-rift succession, developed in the pre-existing extensional basin unit, is thicker than its pre- and syn-inversion sequences part of the post-rift cover. In contrast, inversion anticlines developed in the so far underappreciated Mode II inversion have the opposite syn- versus post-rift succession ratio.

Finally, we offer a brief overview of the multi-faceted impact of structural inversion on the petroleum systems as it is of paramount importance for hydrocarbon prospecting. In our experience, as to finding the optimum zone of inversion for

hydrocarbon prospecting, the proper quantification of the inversion ratio tends to be a challenge. This may be due to a combination of factors such as a) poor seismic imaging of the deeper section beneath an inversion anticline or b) simply not having enough reflection seismic and/or well data available to determine *all* the geometric parameters necessary for calculating the inversion ratio. Therefore a more practical approach is needed to quantify the inversion degree so it could be used in a predictive manner in petroleum exploration.


**2 An unpublished cartoon on inversion tectonics by Albert W. Bally**

The first generalized description of structural inversion was offered by Bally (1984) using a cartoon depicting the evolution of an extensional half-graben subjected to subsequent contraction in three steps. Interestingly, we have found an unpublished extended version of Bally's original inversion model which he designed during the early 2000s at Rice University to show the progression of inversion into the formation of an incipient folded belt (Fig. 1). Specifically, he made this cartoon to illustrate the development of the western Atlas in Morocco. Whereas the inversion at the western termination of the Atlas system has already been described by Hafid et al. (2006), we felt that it is proper to reproduce in this Special Issue of Solid Earth Bally's own schematic summary of inversion tectonics which was left out from that paper (Fig. 1). Importantly, in this unpublished version he added some new elements to his original visual summary (Bally, 1984). In the description of the cartoon summary below we also use some of his unpublished text.

The term inversion tectonics should in our view be restricted to situations where extensional and/or transtensional systems are inverted to form inversion anticlines. Figure 1 attempts to sum up the main characteristics of these systems. During the extensional phase, stratal geometries vary between two end-members depending on the relative rates of sedimentation versus horizontal extension (Fig. 1b and c). Rates of sedimentation which keep up with the extension lead to the familiar growth pattern in the half-graben with characteristic updip convergent strata. In contrast, when sedimentation rates lag behind the extension rates it will result in the subhorizontal infill of the half-grabens.

Minor (or mini) inversions involve partial inversion of the graben fill (Fig. 1d) which could be difficult to differentiate from forced folds (e.g. Withjack et al., 1990) associated with extensional tectonics. Using the simple criteria, i.e. forced folds are monoclines, with one side below regional, whereas inversion structures are anticlines that are above regional, these structures can be distinguished (Mark Rowan, personal communication, 2020). The inversion can proceed until the extensional system is restored to its pre-kinematic configuration, reaching the null-point (Fig. 1e) s*ensu* Williams et al. (1989). As inversion further advances, essentially co-planar reverse faults and/or short-cut faults may form (Fig. 1f). Minor subsidiary décollement systems will eventually appear on the flanks of the uplifts as the stresses are transmitted along the competent strata of the foreland adjacent to both sides of the inversion system.

On seismic reflection profiles inversion structures are characterized by thick asymmetrical anticlinal cores representing the extensional regime with a short, steeper forelimb and a longer, gentler, more planar back limb. The overlying strata thinning onto the structure on both flanks represent the inversion regime. The later updip convergence sequence is most prominent over the maximum graben fill (Fig. 1e) and thus differs from the updip convergence associated with the earlier extensional phase (Fig. 1b). The inversion sequence provides timing constraints for the inversion phase but because of its high position on the structure it is frequently eroded (Fig. 1f).

## 3 A worldwide data base on petroleum fields: reported cases of inversion tectonics

To determine to what degree structural inversion is understood and recognized in the process of exploration, development and production of hydrocarbons we decided to conduct a data mining exercise. In petroleum industry practice, the traps of producing reservoir units in hydrocarbon fields are always specifically described as it provides critical information.

We had access to a very large, almost worldwide (excluding onshore US and Canada) comprehensive data base on hydrocarbon fields and discoveries (IHS Markit, 2020). This data base differentiates between the "trap form type" described in any given field (like "reverse fault" or "thrust") and the "trap forming mechanism" (like "compression" or "inversion"). Obviously, the trap form type is a simpler, observational category compared to the trap forming mechanism which is a more complex, interpretational category.

A query for the word "inversion" under the trap forming mechanism in this huge worldwide data base (IHS Markit, 2020), containing detailed information on about 31,000 fields and discoveries with about 70,000 reservoir units, provided 720 field and about 2,000 reservoir unit matches. Interestingly, this means that only about 2.3% of the fields were classified under the "trap forming mechanism" as inversion. On the level of individual reservoir units within all the hydrocarbon fields worldwide, the corresponding number is 1.7%.

Another data base query on circa 2,000 reservoirs, worldwide (excluding onshore US and Canada) classified as "reverse fault" or "thrust fault" in the trap form type provided only about 60 matches for inversion as a trap forming mechanism. This translates to about 3%, again, a very low proportion of the reported cases (Fig. 2). We believe that these low percentages for reported inversion tectonics in hydrocarbon fields and their reservoir units might be related to the fact that detailed trap descriptions are always difficult to obtain from the operators of these fields. Moreover, the relevant information for the correct classification and reporting is rarely available in publications.

Therefore we believe that during the life-cycle of many exploration, appraisal, development and production projects the term "inversion" is often used quite loosely during the initial exploration and appraisal phase. In contrast, during the development and production phase the exact meaning of inversion as a trap forming mechanism many times becomes irrelevant and it is replaced in the reporting practice by the more generic "compression" or "overthrusting" descriptors. With other words, whereas inversion tectonics appears to be somewhat overstated in exploration, it is quite possibly underreported in production projects.

Another important consideration is the fact that structural inversion has become well known only since the 1980s. Therefore, the traps of many hydrocarbon fields discovered before have been already classified as the result of simple compression or overthrusting. One of the examples we discuss below, a large onshore oil and gas field discovered in Hungary in 1940, clearly illustrates this situation. The trap of the Lovászi field has been traditionally described as a compressional anticline in keeping with the structural observations made long before the advent of inversion tectonics.

**4 Case studies from the Pannonian Basin, Hungary: the Budafa and Lovászi oil and gas fields**

In the basin classification scheme of Bally and Snelson (1980) the Pannonian Basin is the proto type for back-arc basins where extension did not advance to the opening of an oceanic basin. Large, but subtle surface anticlines were known in the western Pannonian Basin for a long time (Fig. 3), referred to as the "Sava Folds" after a local river in the border zone between Slovenia, Croatia and Hungary (Stille, 1924). Based on the regional compilation of vertically exaggerated composite seismic lines in this area, Tari (1994) found many cases for very young, in many cases ongoing, uplift of the pre-

Cenozoic basement from below the Neogene basin fill. This upwarping occurs on different wavelengths (Fig. 4) and it is still ongoing (Ruszkiczay-Rüdiger et al., 2005, in press). The small features include local folding and/or thrusting of the post-tectonic cover with the inversion of syn-rift structures. Whereas these Sava Folds have map-view dimensions on the scale of 10s of kilometers, there are much larger scale neotectonic uplifts in the Pannonian Basin (Fig. 4). In particular, the Transdanubian Range of western Hungary experienced significant neotectonic uplift and exhumation during the Quaternary,

with a wavelength of about 100 km. Tari (1994) distinguished these two end-member categories of uplift with different wavelength and, for the first time, attributed them to regional-scale and local-scale inversion tectonics. Especially, the map-view of the smaller-scale inverted structures (Fig. 3) suggests that the inversion and uplift are propagating into the intra-Carpathian region from the west (Tari, 1994). Indeed, in-situ stress measurements showed compressive stresses in the western part of the Pannonian Basin, while tensile stresses were obtained in the eastern part (e.g. Dövényi and Horváth,

1990).

On the scale of the entire Pannonian Basin, the inverted structures concentrated at the western margin of the basin are gradually propagating eastwards into the basin since the Late Pliocene. Both the regional-scale and local-scale inversions are driven by the ongoing shortening in the broader area, including the Alps, between the Adriatic promontory of Africa and the European plate (Tari, 1994; Horváth, 1995; Horváth and Tari, 1999; Bada et al., 1999, 2007; Tomljenovic and Csontos,

2001; Vrabec and Fodor, 2006; Horváth et al., 2006). Indeed, borehole breakout data indicate a N-S oriented $\sigma 1$ with $\sigma 3$ also being horizontal corresponding to a transpressional strike-slip regime (Tóth and Tari, 2014). Whereas the local-scale inversions are attributed to the reactivation of pre-existing extensional normal faults in an intra-plate compressive stress field, the basin-scale inversions (Fig. 3) are due to large-scale positive deflection of the lithosphere reacting to the build-up of the same stress field (Horváth and Cloetingh, 1996; Cloetingh et al., 2006). As another contributing factor to neotectonic

inversion, Bada et al. (2001) analysed the role of topography-induced gravitational stress in basin inversion in the Pannonian Basin. They found that the kinematics of the inversion of the western Pannonian Basin is consistent with topography-induced gravitational stress which locally exceeds the magnitude of the far-field stress (Bada et al., 2001).

**4. 1 The Budafa field**

The Sava Folds offered obvious drilling targets for hydrocarbon exploration in the first half of the 1900s (e.g. Dank, 1985). The breakthrough came in 1937 when the Budafa anticline was drilled (Fig. 5) as it was the first significant discovery during

the hydrocarbon prospecting efforts in post-war Hungary (e.g. Tari and Berczi, 2018). The Upper Pliocene (Pannonian) beds outcrop on the surface with dips between 3-10° defining an anticlinal four-way closure of about 18-20 km$^2$. Subsequent appraisal and development drilling established the multiple oil and gas reservoir units within the lower Pannonian part of an east-west striking folded anticline at 900 to 1300 m depth. The 2D vintage reflection seismic illustration of the Budafa field (Fig. 5) is oriented perpendicular to the fold axis. It shows the asymmetric nature of the anticline suggesting an underlying master fault on the northern flank of the structure. Even on this moderate quality vintage line acquired in the early 1980s one can interpret the thickening of the Upper-Middle Miocene ("Sarmatian-Tortonian-Helvetian") strata beneath the apex of the anticline providing evidence for the latest Pliocene to Quaternary inversion of a Miocene syn-rift graben. However, the seismic data quality is not good enough to properly delineate the position of the master fault, let alone that of the null-point.

## 4. 2 The Lovászi field

We chose another example of the Sava Folds which provided an important oil and gas find in the region. One of the first major oil fields discovered in Hungary, Lovászi, is also an inversion anticline delineated by potential field data and surface dip measurements in the western Pannonian Basin in 1940. As this particular exploration play was relatively simple, i.e. E-W trending anticlines with relatively shallow Pliocene to Miocene clastic reservoir targets, all the prospects of this play were drilled up as early as in the 1940's (Dank, 1985) and most of them are essentially depleted by now.

Based on abundant well control, the Pliocene to Miocene succession in the broader area was studied by Juhász (1994, 1998). Her sub-regional lithostratigraphic transect, crossing the Lovászi field (Fig. 6), clearly shows a prominent surface anticline with a vertical relief of about 800-1000 m the Pliocene to Upper Miocene (Pannonian; Sarmatian to Badenian) strata compared to their regional levels in this part of the Pannonian Basin. The Lovászi anticline is depicted as a slightly asymmetric one, therefore, in our interpretation, suggesting an underlying master syn-rift fault on its southern flank (Fig. 6). However, given the lack of deep wells penetrating the entire syn-rift core of the inversion anticline, the geometry of the inferred master fault and the location of a null-point along it cannot be established using well data only.

As there is modern 3D seismic data available covering the entire Lovászi field (Tóth and Tari, 2014) the structural history of this anticline can be studied in the context of its inverted nature (Fig. 7). The interpretation of the seismic data (Fig. 7a) reveals the growth of an anticline in the manner depicted in Bally's cartoon (Fig. 1). In particular, the thickening/thinning geometries within the Miocene (Badenian) to Upper Pliocene (Pannonian) strata in the apex of the anticline show the switch from extension to compression (Fig. 7a). Interestingly, flattening on multiple seismic horizons demonstrated an early growth episode of the anticline during the early Pannonian already focusing hydrocarbon migration into the structure (Tóth and Tari, 2014). The main period for the formation of the anticline, however, is clearly post-Pannonian as all the Pannonian reservoirs levels are gently folded (Fig. 7c) into low-amplitude 4-way closures (Fig. 7b).

Historical production from the multiple Pannonian reservoirs of the Lovászi field (Fig. 7c) was about 50 mmbbl oil and 230 bcf gas. Current exploration efforts are focusing on the deeper parts of these anticlines where reservoir quality prediction and imaging of viable traps are the main challenges (Tóth and Tari, 2014). As most of these anticlines are the products of

Pannonian (Pliocene) to Quaternary inversion of Middle Miocene syn-rift half-grabens, proper structural understanding of the core of the anticlines is critical for any future exploration efforts.

## 5   Case studies from the East Mediterranean: the Mango, Goliath structures and the Tamar gas field

The Eastern Mediterranean is another region where numerous inversion anticlines have been described for about a century (Fig. 8). These Syrian Arc structures, as named by Krenkel (1925), extend from the Sinai to the Palmyrides with a typical
trend of ENE-WSW to NNE-SSW (e.g. Walley, 1998). These prominent features formed by the inversion of pre-existing Mesozoic extensional structures from late Cretaceous to Oligocene times. Two main phases of folding have been documented. The first one can be dated as an intra-Santonian (early Syrian Arc phase) and the second is dominantly a late Eocene series of events (late Syrian Arc phase).

From an exploration point of view, the Syrian Arc structures are very important. For example, the traps within several
onshore Egyptian hydrocarbon fields are formed by Syrian Arc events (e.g. Dolson, 2003). Also, Middle to Late Cenozoic Syrian Arc style compressional features are present in the deepwater of the Eastern Mediterranean providing the traps for many deepwater discoveries during the last decade (Gardosh et al., 2008; Gardosh and Tannenbaum, 2014).

It is important to emphasize, that not all Syrian Arc anticlines are basement-involved structures and, therefore, not all of them are inverted features in the strict sense of the word (e.g. Cooper et al., 1989). A regional Upper Triassic salt sequence
provided an effective detachment surface for numerous anticlines in the Damascene segment of the Arc in Syria (Wood, 2015). However, the involvement of the basement in the Palmyrides cannot be excluded for every anticline so there may be a case for decoupled inversion due to the influence of salt.

In northern Egypt (Fig. 8), sedimentation during the Late Cretaceous was interrupted during the Santonian by the development of inversion-related folds (Moustafa, 1988; Sultan and Halim, 1988; Guiraud and Bosworth, 1997; Bosworth et
al., 1999, 2008). The Egyptian segment of the Syrian Arc extends from the Western Desert to the Northern Sinai. The Syrian Arc inversion anticlines have been described from the subsurface and using outcrop studies in the western Desert (Yousef et al., 2019), west of Cairo (Moustafa, 1988), in the Eastern Desert (Moustafa and Khalil, 1995) and in the northern Sinai (Moustafa and Khalil, 1989; Yousef et al., 2010). The formation of the Syrian-Arc has been attributed by Guiraud and Bosworth (1997) to changes in the Africa–Arabia plate motion with respect to the Eurasian plate at the end of the Santonian.
In Israel and Palestine, mostly subsurface data suggest that many of the Syrian Arc structures are in fact associated with the reverse reactivation of pre-existing Triassic and Jurassic extensional faults (Freund et al., 1975; Druckman et al., 1995). Contraction started in the latest Cretaceous and continued through the Neogene (Eyal and Reches, 1983; Eyal, 1996; Walley, 1998). Just like on the regional scale, two periods of inversion tectonics were documented using reflection seismic and well data. An earlier phase of Senonian to Eocene inversion was followed by a later phase in the Miocene (Gardosh and
Druckman, 2006). Interestingly, early Syrian Arc phase (Syrian Arc I) inverted structures are mostly located onshore and in the narrow shelf area of the Levantine Basin (Figure 8). Most of these thrust fault controlled structures are asymmetric

anticlines with high-amplitude and short wave-length (Gardosh and Tannenbaum, 2014). In contrast, the inversion anticlines of the late Syrian Arc phase (Syrian Arc II) are found in the offshore part the Levantine Basin (Fig. 8). These structures are subtle, having a generally low-amplitude, but large map-view closure (Gardosh and Tannenbaum, 2014). They also appear as just subtly asymmetric folds which were still active during the Messinian (see later). Their inversional origin may not be obvious given the fact that the underlying Paleozoic(?)-Mesozoic extensional structural fabric is typically poorly imaged in the deepwater Levant Basin due to the great depth. However, the better imaged inboard Syrian Arc I anticlines (Fig. 8) provide very good structural analogues (Gardosh et al., 2008).

## 5.1 Inverted "Syrian Arc I" structures offshore Egypt: the Mango and Goliath structures

As subsurface examples of a typical Syrian Arc inversion anticline, we chose two offshore Sinai structures in Egypt (Fig. 8). The Mango structure (Fig. 9) is a circa 24 km long, WSW-ENE oriented anticline (Yousef et al., 2010). After an initial oil discovery drilled on the structure in 1986, two more appraisal wells have been drilled to test the potential of the Lower Cretaceous clastic sequence in this inverted structure. The thickening-thinning relationships within the Mesozoic-Cenozoic sequence define a Jurassic to Early Cretaceous period of normal faulting along a poorly imaged master fault (Fig. 9). In contrast, the overlying mostly Cenozoic sequence displays progressive onlap onto the apex of the anticline.

The Goliath structure (Fig. 10) is a circa 20 km long, SW-NE oriented anticline with steeper dips on its northwestern flank (Ayyad et al., 1998). The Goliath-1 well was drilled in 1996 targeting the Lower Cretaceous clastic sequence which had multiple oil reservoirs in the nearby Mango inversion structure, about 15 km to the WSW (Fig. 8). The well turned out to be dry but penetrated the Lower Cretaceous down to 3200 m. The thickening-thinning relationships within the Mesozoic-Cenozoic sequence define a Jurassic(?) to Early Cretaceous period of extensional faulting along a master fault (Fig. 10). The overlying mostly Cenozoic sequence here also displays progressive onlap onto the apex of the anticline. Both the Mango and Goliath doubly plunging anticlines belong to the many other Syrian Arc I inverted structures in the offshore North Sinai with a characteristic ENE to NE strike (Fig. 8).

## 5.2 An inverted "Syrian Arc II" structure in Israel: the Tamar gas field

The giant Tamar field, with its 7-8 tcf biogenic gas reserves, is located in deepwater Israel (Fig. 9) and it was discovered in 2009 (Needham et al., 2017). On a depth-converted regional seismic section (Fig. 11a) the Tamar structure is a very prominent anticlinal feature mostly, but not entirely pre-dating the overlying Messinian evaporite sequence. In map-view (Fig. 11b) the Tamar anticline has a slightly asymmetric closure trending SW-NE. The prominent NW-SE striking "piano-key" faults (Kosi et al., 2012) cross-cutting the anticline are quite typical for the entire deepwater Levant Basin. These faults are not sealing in nature as the GWC is at the same depth of 4797 m (Fig. 11b) across the entire Tamar gas field (Needham et al., 2017). A dip-oriented section we have constructed across the Tamar field assumes the isopachous nature of the three main reservoir intervals (Sand A, B and C) reported by Needham et al. (2017) and it reveals the subtle asymmetry of the structure (Fig. 11c). The slightly steeper SE flank of the anticline suggests an underlying master fault, yet, the regional-scale

seismic section (Fig. 11a) does not display such a fault anywhere down within the Cenozoic sequence. Similarly to the
Lovászi field discussed earlier (Fig. 7), the reservoirs of the Tamar field are located stratigraphically fairly high within the
structure, in the post-rift sequence. These Miocene reservoirs are located in an isopachous sequence which was deposited
before the inversion took place (Fig. 11c). Since the Mesozoic (Jurassic-Early Cretaceous?) master fault responsible for the
inversion is located a few kilometers beneath the Tamar anticline its geometry is poorly defined by the seismic data (Fig.
11a). The existence of a Mesozoic syn-rift graben at depth is mostly supported by the analogy with the much better imaged
and understood Syrian Arc structures located closer to the coastline (Gardosh et al., 2008; 2010, 2011; Gardosh and
Tannenbaum, 2014).

## 6  Case study from Atlantic Morocco: offshore anticlines of the Atlas system

Inversion tectonics in the Atlas Mountains of Morocco has been first described in the 1990s by Laville and Piqué (1992)
Giese and Jacobshagen (1992); Lowell (1995) and Beauchamp et al. (1996, 1999). For the last two decades many other
publications were devoted to various aspects of inversion in the broader Moroccan Atlas system (Frizon de Lamotte et al.,
2000, 2009; Hafid, 2000, 2006; Teixell et al., 2003, Missenard et al., 2007; Leprêtre et al., 2018; Perez et al., 2019).
The Atlas sytem does not stop at the Atlantic coastline, but the large surface anticlines of the onshore onshore Essaouira
Basin and the western High Atlas of Morocco  (Fig. 12) can be followed into the nearby offshore area (Hafid, 2000; Hafid et
al., 2000; Hafid, 2006; Hafid et al., 2006). In fact, the signature of the Neogene to Recent inversion of the Atlas system, as
the result of African-Eurasian plate convergence, can be followed into the deepwater area as well, some 200 km to the west
from the coastline in water depth of 2,000-4,000 m (Fig. 12), but still located over highly extended continental crust (Tari
and Jabour, 2011; 2013; Tari et al., 2012; Neumaier et al., 2016). The deepwater anticlines have a general NW-SE to WNW-
ESE trend based on regional-scale 2D seismic reflection data sets (Fig. 12b).
These inversion anticlines, with a an average spacing of about 10-20 km, are superposed by a much longer wavelength
neotectonic arching of the Ras Tafelney Plateau (Fig. 13) described by Tari et al. (2012). The reason for this regional, about
200-300 km wide basin-scale inversion of the margin (Fig. 13) is the neotectonic shortening within the broader African–
Eurasian plate-boundary (Gomez et al., 2000) extending at least 200 km offshore from the Atlas system onshore (Fig. 12).
The broad deepwater arch was termed the "Atlantic Atlas" by Benabdellouahed et al. (2017) and we consider it as a good
example of basin-scale inversion.
Note that most of the prominent inversion anticlines in the onshore Essaouira Basin and the High Atlas region (Fig. 12a)
have been interpreted in terms of folding detached on Upper Triassic to Lowermost Jurassic evaporites in the region (e.g.
Hafid, 2000, 2006; Verges et al., 2017; Dooley and Hudec, 2020). However, the anticlines in the deepwater area, outboard of
the salt basin, are entirely controlled by basement involved faults (Tari et al., 2012). Based on the interpretation of 3D
reflection seismic data, compressionally reactivated syn-rift normal faults are responsible for these inversion structures with

a corresponding detachment close to the base of the Mesozoic basin fill (Fig. 14a). This kind of inversion anticline was termed as basement–involved inversion fold (Fig. 14b) by McClay et al. (2018) quoting Mount et al. (2004).

The example shown in Fig. 14a forms a robust 4-way closure as the result of inversion tectonics, like many others along the basinward edge of the salt basin (Fig. 13), making it a hydrocarbon exploration target. The bright reflectors within the mid-Jurassic sequence (Fig. 14a) were interpreted corresponding to a basin floor fan (Fig. 14a) and this exploration target was drilled in 2014 (Tari et al., 2017b). The Mazagan-1 (MZ-1) well turned out to be a dry hole, for reasons other than finding a valid hydrocarbon trap.

**7 The impact of inversion tectonics on petroleum systems and exploration efforts**

Whereas numerous publications are devoted to the structural geology of inversion, there are only a few papers which tried to generalize the impact of inversion tectonics on petroleum systems (e.g. Macgregor, 1995; Turner and Williams, 2004; Cooper and Warren, 2010, 2020; Bevan and Moustafa, 2012). Looking at Bally's cartoon (Fig. 1) it is intuitive to assume that there has to be an optimum "Goldilocks Zone" of inversion tectonics from a petroleum exploration point of view. However, this optimum is not simply the function of the trap size but also the function of the complex interaction between the source, reservoir and seal rocks via hydrocarbon migration.

From a strictly petroleum systems point of view the positive connotation of inversion tectonics in the petroleum industry is largely due to the trap and the reservoir development, i.e. robust closures with reservoirs in them as the syn-rift basin fill tends to accumulate reservoirs. In contrast, the negative connotation of inversion tectonics is based mostly on its perceived impact on charging and sealing, i.e. the uplift shuts down generation in the syn-rift source kitchen and the ongoing deformation tends to lead to breaching and exhumation.

The summary below is largely based on the work of Bevan and Moustafa (2012) who used the examples of three onshore Egyptian fields (e.g. Razzak, Mubarak and Kattaniya) to generalize some observations. We note that these cases specifically capture the learnings from inverted structures in a failed wide rift setting in an onshore basin where the post-rift basin fill is very thin, especially compared to the syn-rift sequence (Fig. 15).

Inversion structures which are relatively mild develop low-amplitude but robust 4-way closures in the hangingwall of the master fault responsible for the structure (Fig. 15a). The master fault does not necessarily have to manifest itself at the level of the reservoirs. As described earlier in the case of the Lovászi and Tamar fields, inverted structures could have large closures higher up in the unfaulted sequence (Figs. 7 and 11, respectively). As to charge, the position of source and seal rocks in the hangingwall side of the fault is quite critical. Whereas the hydrocarbons generated in the source rocks located deep beneath the inversion anticline may migrate updip towards the flexural margin from the structure, the source rocks at the faulted margin may generate hydrocarbons which then migrate up along the fault plane to the ultimate trap in the apex (Fig. 15a).

In the more advanced inverted structure (Fig. 15b) the same basic charge limitation occurs, i.e. the majority of mature hydrocarbons from within the source rocks within the deeper syn-rift sequence will migrate away from the hangingwall closure associated with the reactivation of the master fault. However, the smaller closures that could develop above antithetic faults, or detached on salt on the subsidiary side of the half-graben (Fig. 1e) may receive charge (Fig. 15b). This asymmetric arrangement of traps associated with near null-point inversion is informally called the butterfly structure (see Fig. 1e).

In the most advanced cases of structural inversions (Fig. 15c), the reservoir units in the hangingwall become uptilted and potentially exposed on the paleo-surface, therefore becoming breached. As noted by many, the vertical uplift of source rocks, potentially generating hydrocarbons prior to the inversion, may switch off the kitchen as the source rocks may reach shallower depth where they are not generating any more (e.g. Turner and Williams, 2004; Cooper and Warren, 2010). In these more severe cases of inversion, the smaller, subsidiary structures on the flank should be targeted (Fig. 15c). These smaller closures may remain unbreached and could receive charge from downdip source rocks as Bevan and Moustafa (2012) pointed out.

The Western Desert also provides important other insights into the positive aspects of inversion. Bosworth and Tari (2020) describe a case study where intervening periods of inversion in multi-phase extensional basins turned out to be a key to have excellent exploration targets.

As to modelling the hydrocarbon charge history in inversion structures, especially those associated with multi-step deformations, it is clearly a difficult 4D challenge. In a case study of the Wytch Farm field Neumaier et al. (2017) were able to explain the exceptional oil charge in this large field as opposed to many other adjacent inversion structures. However, for this modelling, numerous steps are required which are typically not parts of basin modelling efforts on more "standard" fields. These extra modelling steps included pre-inversion drainage area-based oil migration analysis, petroleum systems modeling of the duration of the localized "charge window", cross-fault charge amounts and rates, etc.

Another unique phenomenon associated with inversion is the fault-valve action described by Sibson (1995). The selective reactivation of pre-existing nornal faults as reverse faults could be due to fluid overpressures developed during inversion. The significant overpressure can trigger the compressional reactivation of moderately to steeply dipping faults. Therefore, episodes of vertical hydrocarbon remigration of hydrocarbons from lower reservoir levels to higher ones are quite likely to occur along the reverse faults bounding the inversion structure (Sibson, 1995). In an extreme case, however, this repeated process can breach an existing petroleum accumulation.

As to the regional scale impact of structural inversion we reiterate here the simple point made by Tari and Jabour (2011). The large gas discoveries of the last decade in the deepwater Levant Basin are all associated with inverted structures which strike parallel to the margin (Fig. 8). From a trapping point of view this translates to an optimum situation as the closures of the four-way anticlines are not significantly affected by the regional basinward dip trending perpendicular to the anticlinal axes (Fig. 16). In contrast, on passive margins where the inversion anticlines have the same trend as the regional dip the four-way closures on the updip end of the structures tend to be much smaller (Fig. 16). The anticlines in the central segment of Atlantic Morocco have a general WNW-ESE trend (Fig. 12), perpendicular to the overall strike of the passive margin, but

parallel with the regional dip of the margin. Therefore we believe that the regional-scale trend of the inverted structures versus the regional dip in a passive continental margin or in a foredeep setting is quite important (Tari and Jabour, 2011).

## 8 Two modes of inversion tectonics: petroleum exploration implications

Whereas inversion tectonics can produce spectacular traps, inversion tectonics is a process which has profound implications on other elements of the petroleum systems and, therefore, the hydrocarbon prospectivity, both in a positive and a negative
sense (e.g. Macgregor, 1995; Turner and Williams, 2004; Cooper and Warren, 2010, 2020). The most negative impact is attributed to the fact that during inversion source-rock sections are brought closer to the paleo-surface and therefore previous mature source-rocks switch-off and become non-generative. Also, the main reservoir and source-rock sections are brought to the near surface and therefore breached (Fig. 17a). There are many other negative, but valid impacts listed by Turner and Williams (2004) giving the impression that inverted features may be more challenging for exploration than "regular"
anticlines formed by simple contraction. Perhaps their view might also be somewhat biased by considering examples from exhumed European Atlantic margins (e.g. Doré et al., 2002). In these regionally inverted rift basins there is plenty of evidence for underfilled fields and former petroleum accumulations which were breached and leaked away due to inversion tectonics (Turner and Williams, 2004). In particular, the Wessex Basin (Underhill and and Stoneley, 1998) provides a very well studied example of this process. As Neumaier et al (2017) demonstrated, the petroleum charge in this basin has been
successful only for the very large Wytch Farm and a series smaller satellite traps along a fill/spill chain to the west, but did not work anywhere else.

Yet, in many other basins of the world, inverted structures provide repeatable and highly successful plays. In particular, some of the examples we chose for this paper, located in the Sava Folds region of the western Pannonian Basin and the Syrian Arc II anticlines in the deepwater Levant Basin turned out to be very successful.
We believe that the key for the success in these basins is that source rocks are not constrained to the extensional basin fill beneath but rather occupy a higher and broader post-rift, but pre-inversion stratigraphic position (Fig. 17b). These post-rift source rocks tend to be more regional in character and can have the right depth within the hydrocarbon generation window as opposed to the much deeper syn-rift source rocks which are spent and cannot expel anymore. Given the position of the active source rock sequence in the lower Miocene to Oligocene post-rift basin fill in the Tamar field example (Gardosh and
Tannenbaum, 2014), the hydrocarbon generation could be assumed regionally and the inversion anticlines becomes the focus of the ongoing lateral and vertical charge. These structures can therefore be more successful at shallower reservoir levels within the post-rift succession.

Due to the observational evidence, we suggest that examples of inversion tectonics represent two main modes of evolution (Fig. 17). Mode I is the more widely recognized "classic" case, where the syn-rift succession is overlain by a relatively thin
post-rift cover including pre-, syn- and post-inversion units (Fig. 17a). This could be result of a relatively short time period between the end of extension and the onset of contraction. Cooper and Warren (2020) using a recently provided evidence for

an early observation by Mike Coward to show that  inversion indeed appears to be favoured by a relatively short time interval between the extensional and compressional phases. Another reason for a pre-existing extensional basin unit that is thicker than its post-rift cover could be low sedimentation rate due to limited accommodation space and/or sediment supply.

In contrast, inversion anticlines developed in the so far underappreciated Mode II have the opposite proportions, i.e. post-rift sequence is thicker than the underlying syn-rift (Fig. 17b). This might be due to a longer time interval between the extension and contraction but it can also reflect high sedimentation rates due to large sediment supply and accommodation space. Importantly, the contractionally reactivated faults do not propagate through the entire post-rift basin fill and therefore Mode II structures manifest themselves typically as buckle folds at higher stratigraphic levels (e.g. Tamar Anticline, Fig. 11).

In general, we tentatively attribute the grouping of inversion structures in these two modes to their regional geodynamic setting. Mode I structures are likely to be associated with failed, intra-continental rifts (e.g. North Sea) and proximal passive margins (e.g. offshore Sinai shelf margin). Some of these structures may have also experienced regional-scale exhumation which prevented prolonged post-rift sedimentation (e.g. Wessex Basin). Mode II structures appear to be found in back-arc basins (e.g. Pannonian Basin) and in distal parts of passive margins (e.g. deepwater Levant Basin). Folded belts and their

foredeeps may have either Mode I or II inversion examples depending on the precursor rift basins incorporated in the subsequent contractional deformation.

## 9 Quantifying inversion: limitations of the existing methods

Finally, we would like to emphasize the need to better quantify the degree of inversion for any given structure in order to find the optimum trapping situation for exploration efforts on a global scale. With other words, what degree of inversion provided the largest number of hydrocarbon fields worldwide? This analysis requires a quantitative description of the inversion and there are two ways of doing this (Fig. 18). Williams et al. (1989) introduced the concept of inversion ratio, i.e. the magnitude of contraction due to inversion versus the magnitude of extension. In seismic profiles, this is equivalent to the

ratio between the thickness of syn-rift deposits above the null point parallel to the fault plane and the total thickness of syn-rift deposits parallel to the fault plane on the hanging-wall (Fig. 18a). However, the inversion ratio may be difficult to calculate in cases when the null-point cannot be located with confidence. An alternative method was proposed by Song (1997) to calculate the inversion ratio (Fig. 18b), but this method also requires good handle on many elements of the stratal geometry along the master fault (e.g. Yang et al., 2011).

In our experience, the quantification of inversion degree is a challenge as the deeper section beneath an inversion anticline is typically not well imaged seismically and/or not drilled due to the greater depth. It is quite typical in exploration projects *not* have all the geometric parameters necessary for calculating the inversion ratio for any given structure (Fig. 18). For example, the inversion ratio cannot be determined with confidence in most of the case studies described in this paper, i.e. Budafa, Lovászi and Tamar structures (Figs. 5, 7 and 11, respectively) due to the poor understanding of the underlying extensional

basins. Therefore a more practical approach is needed to describe inversion tectonics in cases where not all the geometric

elements of a structure can be measured due to subsurface data constraints. This may be especially true for Mode II inversion structures.

## 10 Conclusions

Inversion structures provide a range of traps for petroleum exploration. The most preferred petroleum exploration targets are mild to moderate inversion structures with relatively small vertical amplitude, simple map-view expression. The hydrocarbon traps in these structures cluster above the extensional depocentres closer to the faulted margin and are typically well imaged on seismic reflection data. Cases of strong or multiple inversion are not considered ideal for exploration, mostly due to leakage or breaching. Besides seal failure multiple migration episodes may also result in underfilled traps. As syn-rift source rock units may be uplifted above the generation window the hydrocarbon charge may become reduced or terminated. Severe cases of inversion may also translate to reflection seismic imaging challenges associated with the complex trap(s).

For any particular structure the evidence for inversion is typically provided by subsurface data sets such as reflection seismic and well data. However, in many cases the deeper segments of the structure are either poorly imaged by the seismic data and/or have not been penetrated by exploration wells. In these cases the interpretation of any given structure in terms of inversion has to rely on the regional understanding of the basin evolution with evidence for an early phase of substantial crustal extension by normal faulting. In some cases, where the regional geology has not been properly appreciated, the simple reactivation of pre-existing structures related to earlier episodes of shortening was erroneously classified as inversion. The sometimes negative perception of inversion is due to the fact that there are not that many successful examples described globally. Only about 3% of the traps of hydrocarbon fields with reverse faulting or overthrusting are reported to be associated with inversion (Fig. 2). We believe that this number should be significantly higher as many inverted structures may not be recognized as such. A statistically driven global data-mining approach, establishing observationally the most optimal degree of structural inversion for hydrocarbon exploration, appears to be missing to date.

There might be a negative bias towards the prospectivity of inverted structures using examples from exhumed margins. Another bias may stem from the typical assumption that the generating kitchen tends to be in the syn-rift sequence of the inverted structure. Successful exploration cases in basins which have not experienced uplift and exhumation, like the giant gas discoveries in the deepwater Levant Basin, highlighted the importance of the source, reservoir and seal rocks not being constrained to the syn-rift basin fill. In these cases, all these elements of the petroleum system are located in the regional post-rift pre-inversion sequence.

Inversion structures should be classified in two modes, depending on the relative ratio of the syn-rift versus post-rift strata (up to the stratigraphic level of the latest significant inversion event). Mode I corresponds to the classic inversion structures, i.e. the reverse-fault bounded "Sunda-folds", dominated by a thick syn-rift basin fill with a relatively thin post-rift sequence at the time of inversion. In contrast, Mode II structures develop when inversion occurs after the deposition of a thicker post-

rift sequence than the syn-rift basin fill in the underlying extensional structure. In this case buckle folds tend to develop in the post-rift pre-inversion sequence. Cases of Mode I inversion tend to develop in failed, intra-continental rifts and proximal passive margins and Mode II structures appear to be found in back-arc basins and in distal parts of passive margins.

## Data availability

Some of the seismic lines used in this study are confidential and not available publicly.

## Author contribution

Gabor Tari wrote the text and compiled the manuscript, Didier Arbouille analysed the IHS Markit global data base, Zsolt Schleder provided structural geology expertise and Tamás Tóth supplied 3D reflection seismic and well data from SW Hungary.

## Competing interests

The authors declare that they have no conflict of interest.

## Acknowledgements

We are pleased to thank Jonas Kley and Piotr Krzywiec for inviting this paper to the special issue of Solid Earth and for their editorial efforts. We are grateful to IHS Markit for the access to their extensive worldwide exploration and production data base in order to find reported cases of inverted traps. Discussions about inversion tectonics with Albert Bally, Bill Bosworth, Joan Flinch, Mohammad Hafid, Frank Horváth, Haddou Jabour, Hans-Gert Linzer, Paul Lyon and Juan Ignacio Soto are also acknowledged. Mark Rowan, Bill Bosworth, Michael Gardosh and Gabor Bada provided very detailed, insightful and constructive reviews. Peter Pernegr kindly drafted some of the figures. This paper is dedicated to the memory of Prof. Albert W. Bally (1925-2019) who was the PhD Advisor of the first author at Rice University, Houston, Texas.

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

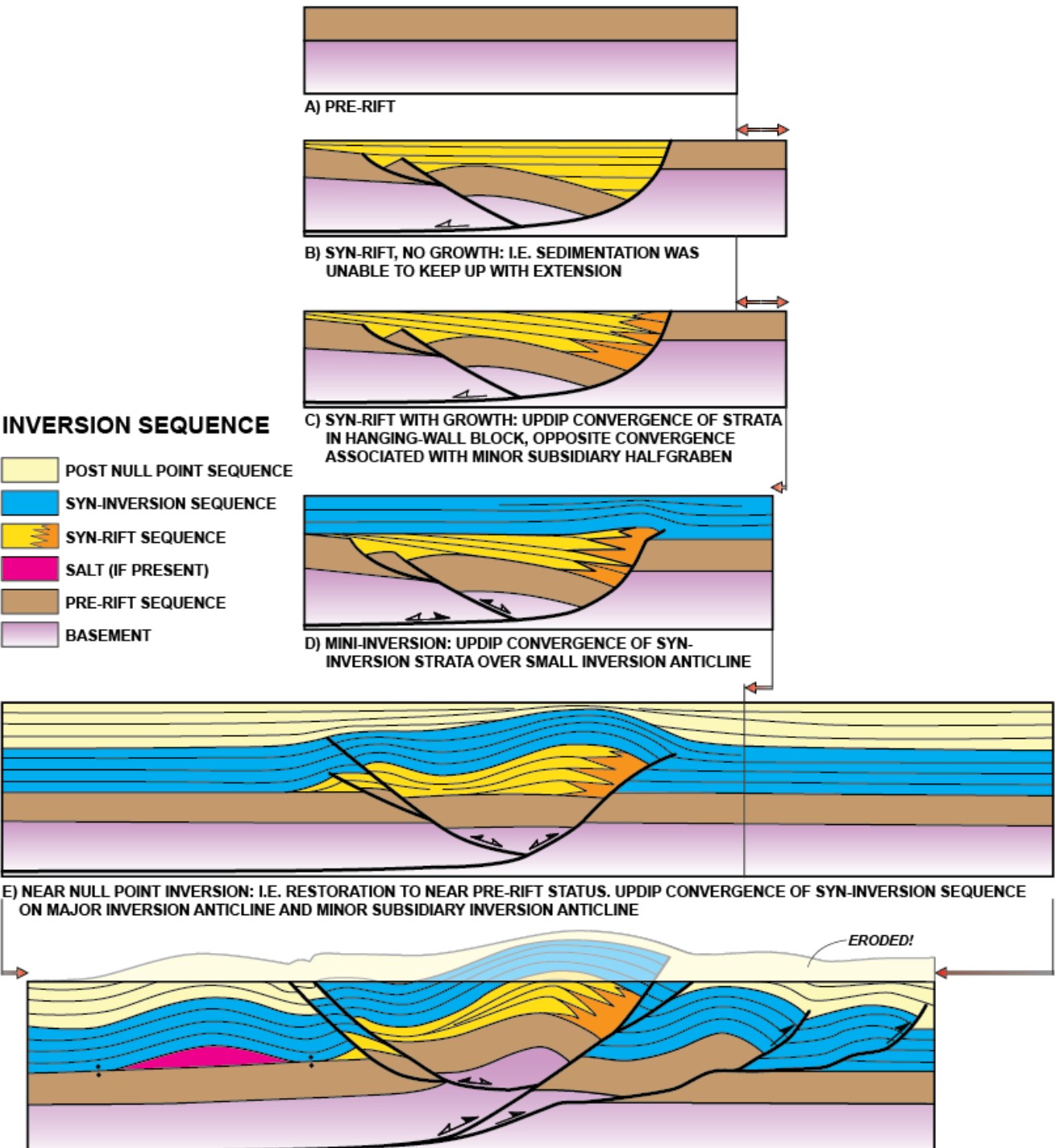

**INVERSION SEQUENCE**

- POST NULL POINT SEQUENCE
- SYN-INVERSION SEQUENCE
- SYN-RIFT SEQUENCE
- SALT (IF PRESENT)
- PRE-RIFT SEQUENCE
- BASEMENT

**A) PRE-RIFT**

**B) SYN-RIFT, NO GROWTH: I.E. SEDIMENTATION WAS UNABLE TO KEEP UP WITH EXTENSION**

**C) SYN-RIFT WITH GROWTH: UPDIP CONVERGENCE OF STRATA IN HANGING-WALL BLOCK, OPPOSITE CONVERGENCE ASSOCIATED WITH MINOR SUBSIDIARY HALFGRABEN**

**D) MINI-INVERSION: UPDIP CONVERGENCE OF SYN-INVERSION STRATA OVER SMALL INVERSION ANTICLINE**

**E) NEAR NULL POINT INVERSION: I.E. RESTORATION TO NEAR PRE-RIFT STATUS. UPDIP CONVERGENCE OF SYN-INVERSION SEQUENCE ON MAJOR INVERSION ANTICLINE AND MINOR SUBSIDIARY INVERSION ANTICLINE**

*ERODED!*

**F) POST NULL POINT CONTRACTION: FORMATION OF BASEMENT-INVOLVED REVERSE FAULTS, SATELLITE DECOLLEMENT FOLDED BELTS AND COMPRESSIONAL SALT ANTICLINES**


**Figure 1: Extended version of Bally's (1984) original inversion model. This cartoon was redrafted after an unpublished figure made by Albert W. Bally in the early 2000s at Rice University to show the progression of inversion tectonics into the formation of an incipient folded belt. Specifically, he made this cartoon with the western high Atlas of Morocco in mind (e.g. Hafid et al., 2006) that is why salt is shown here as a detachment level accommodating some of the contraction.**


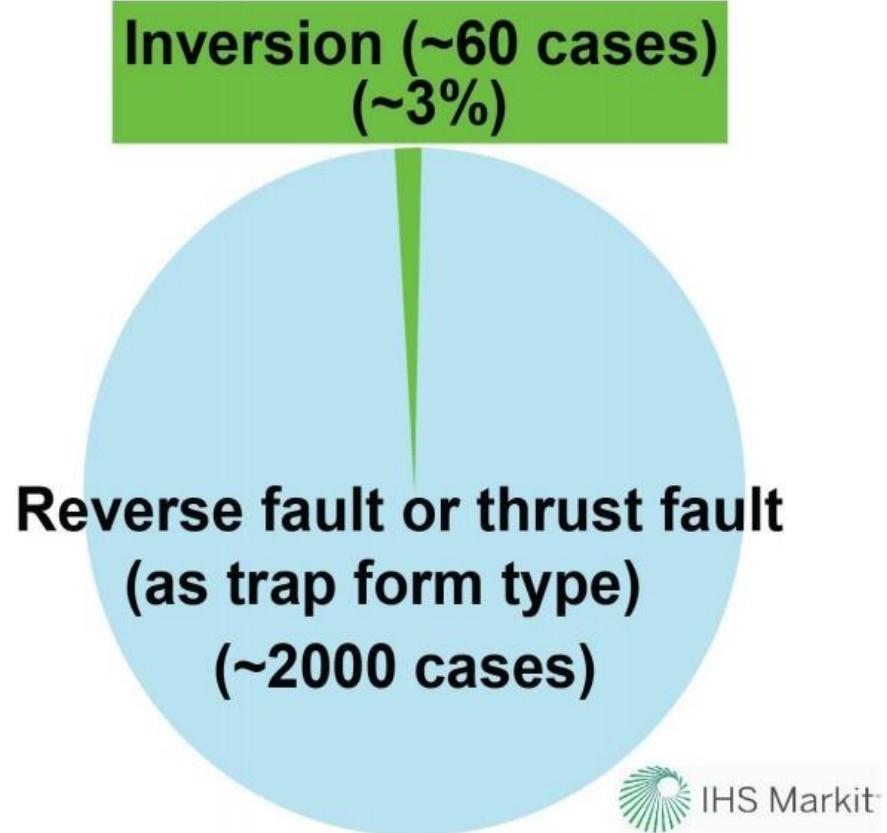

**Figure 2: Outcome of a worldwide (excluding onshore US and Canada) data base search using hydrocarbon fields and discoveries with circa 2,000 reservoir units. In these reservoir units the "trap form" type was classified as "reverse fault" or "thrust fault". Interestingly, within these 2,000 cases we have found only about 60 matches for inversion as a "trap forming mechanism". This**
**translates to only about 3%, a strikingly low proportion. We believe that inversion tectonics may be unrecognized in many fields globally and therefore it remains underreported. Courtesy of IHS Markit.**


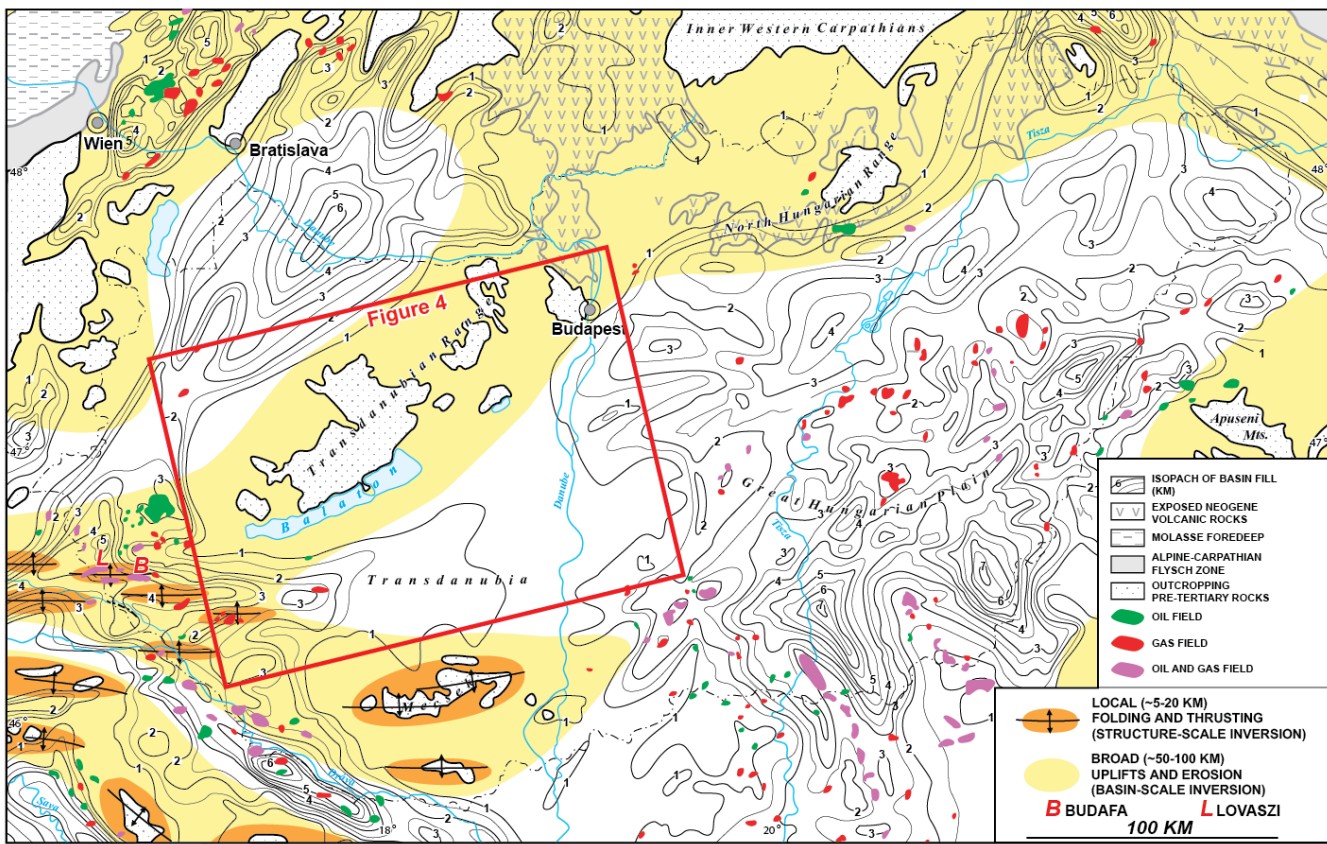

**Figure 3: Regional map of the central part of the Pannonian Basin adapted from Tari (1994) who distinguished for the first time two classes of inversion tectonics in this basin. These two end-member categories of uplift are attributed them to regional-scale and local-scale structural inversion tectonics. The map-view pattern of the smaller-scale inverted structures suggests that inversion and uplift are propagating into Pannonian Basin from the SW driven by the Adria plate. The isopach of the basin fill and the location of hydrocarbon fields are based on Tari and Horvath (2006).**


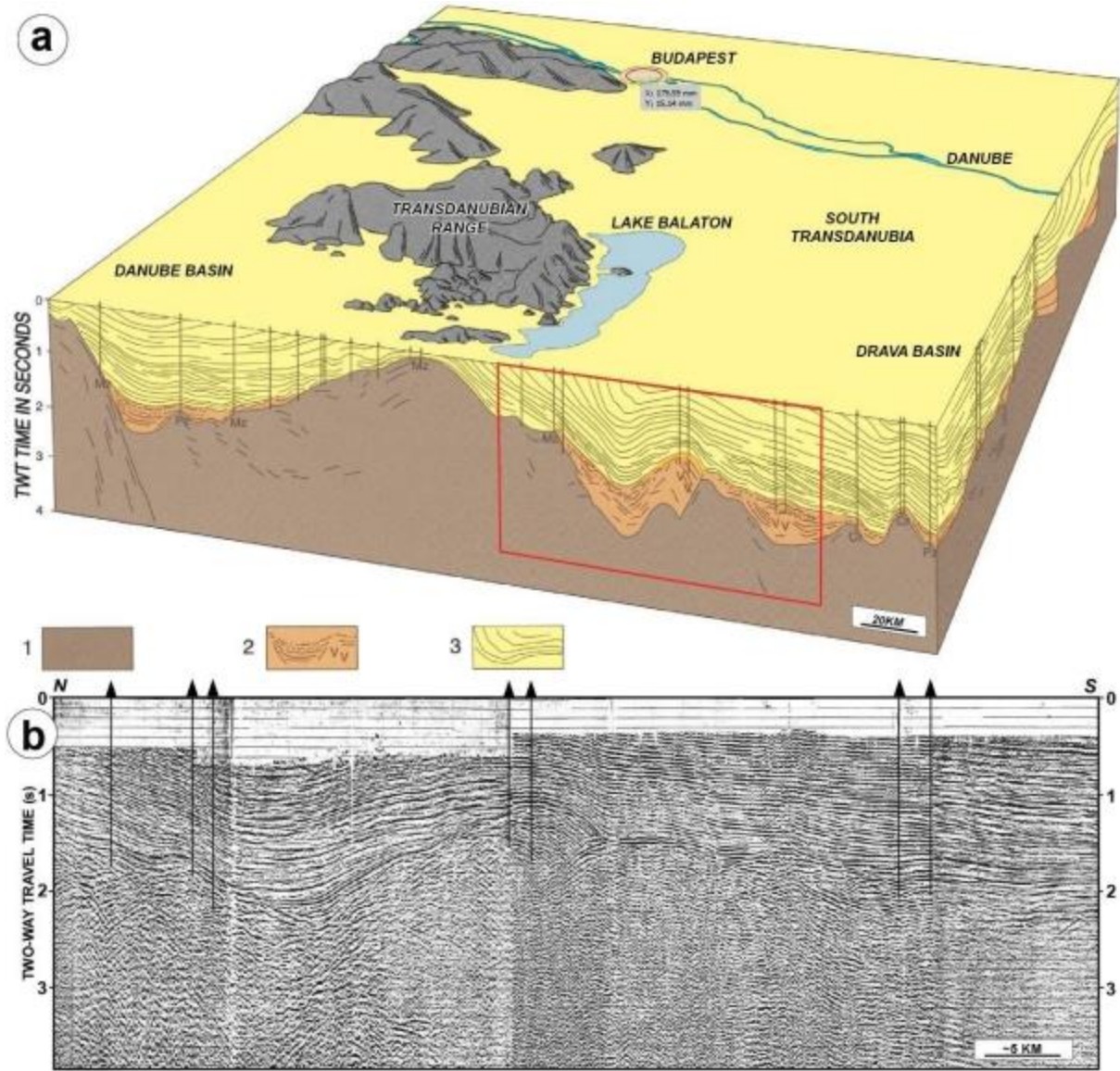


**Figure 4: a) Block diagram of the Transdanubian part of the Pannonian Basin to illustrate the structural and stratigraphic conditions, and their relation to surface morphology. The regional- scale upwarping of pre-Cenozoic rocks of the Transdanubian Range north of the Lake Balaton (for location, see Fig. 3) is the consequence of Pliocene to Recent basin inversion (adapted from Horvath and Tari, 1999). Legend: 1, Mesozoic-Paleozoic bedrocks; 2, mid-Miocene syn- rift strata; 3, late Miocene to Pliocene**
**post-rift strata b) Composite vintage 2D seismic section (Tari, 1994; Rumpler and Horváth, 1988) as part of the regional seismic line shown above as a line drawing. The core of the young anticline in the center was drilled by wells Vése-1 and -2 and penetrated thick syn-rift Miocene strata. The overall geometry of the structure suggests that a syn-rift half-graben was inverted, just like in the case of the Budafa anticline (Fig. 5).**

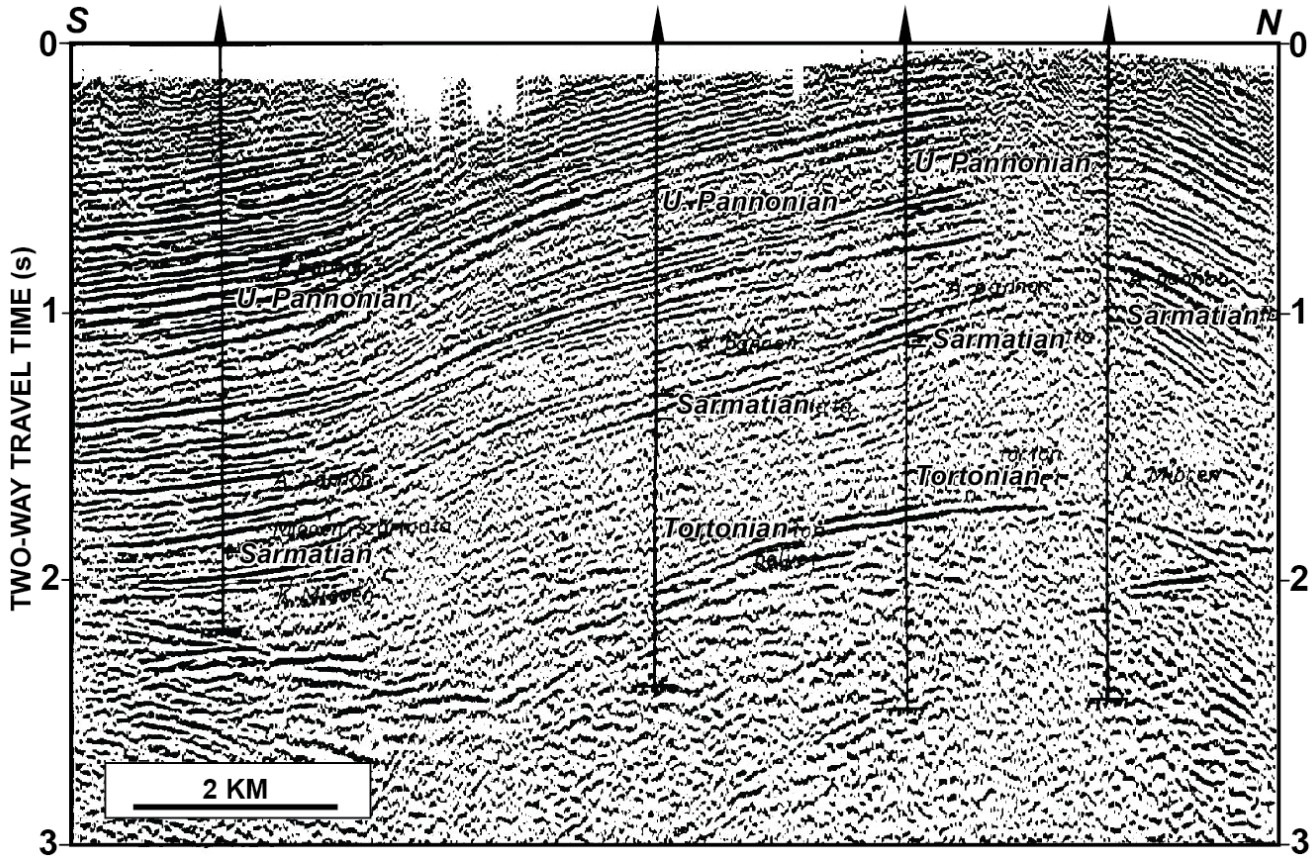

Figure 5: Vintage 2D reflection seismic illustration of the Budafa field (Dank, 1988; Pogacsas et al., 1994). The asymmetric nature of the surface anticline suggests an underlying master fault on the northern flank of the structure. Even on this moderate to poor quality vintage line one can interpret the thickening of the Upper-Middle Miocene ("Pannonian-Sarmatian-Tortonian") strata beneath the apex of the anticline. This provides evidence for the latest Pliocene to Quaternary inversion of a pre-existing Miocene syn-rift half-graben. Unfortunately, the seismic data quality is not good enough to properly delineate the position of the master fault, let alone that of the null-point.


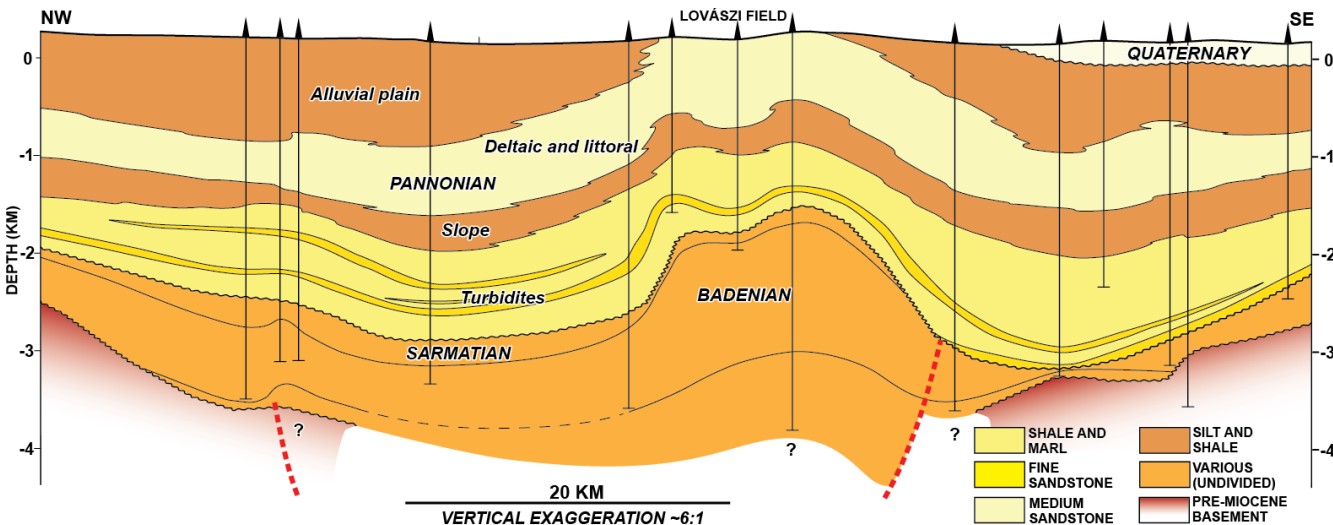

**Figure 6: This sub-regional lithostratigraphic transect, crossing the Lovászi field, was redrafted after Juhász (1994, 1998). Her study was not devoted to the structural evolution of the area but rather to the correlation of various lithologies of the Miocene and Pliocene sequence using primarily well data. Note the missing Pannonian sequence above the Lovászi anticline which could have a 800-1000 m thickness. Given the slightly asymmetric shape of the Lovászi anticline we inferred the presence of a master syn-rift fault and added it to the original illustration (dashed red line) beneath its southern flank (cf. Figure 7).**


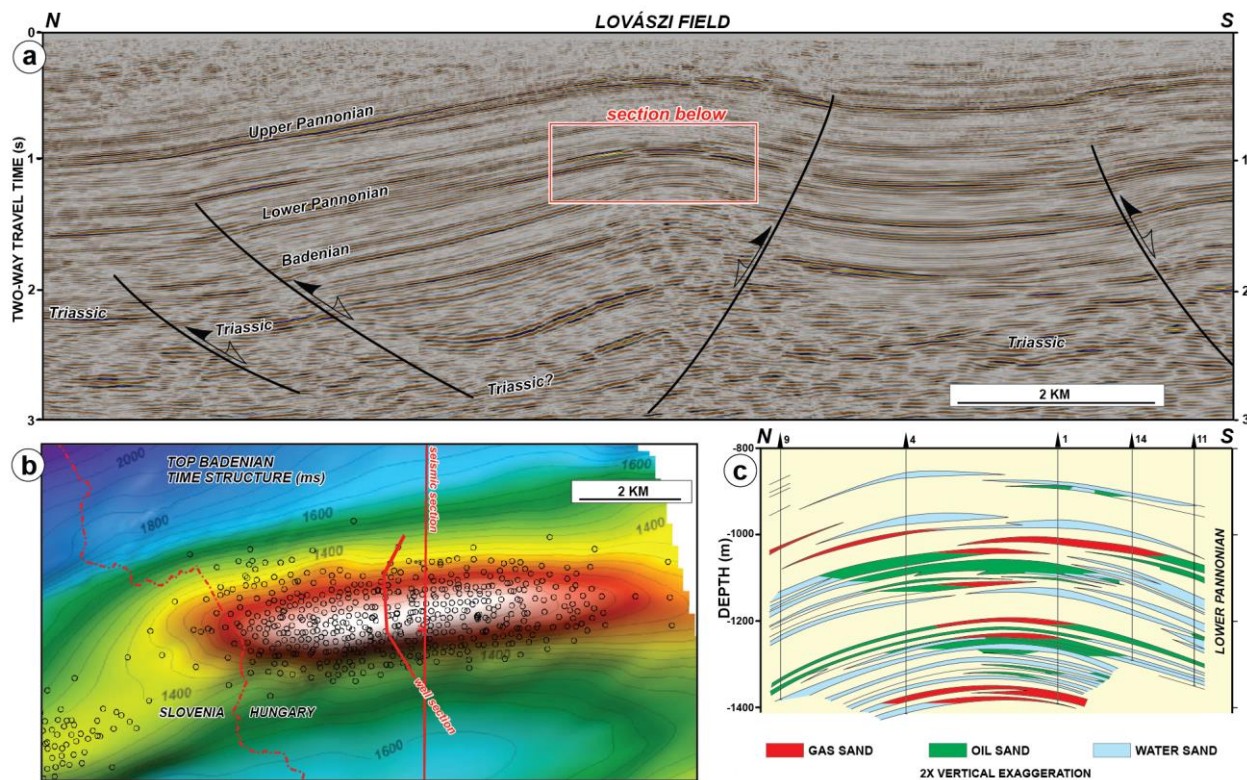

**Figure 7. Highlights of the Lovászi field, at the border of Hungary and Slovenia, for location see Figure 3. a) 3D reflection seismic example across the field adapted from Tóth and Tari (2015) with no vertical exaggeration assuming a 3 km/s average seismic velocity. b) Two-way travel time structural map on the top Badenian (Middle Miocene) seismic horizon with a 25 ms contour interval. c) Well-based cross section across the central part of the field redrawn from Dank (1985).**

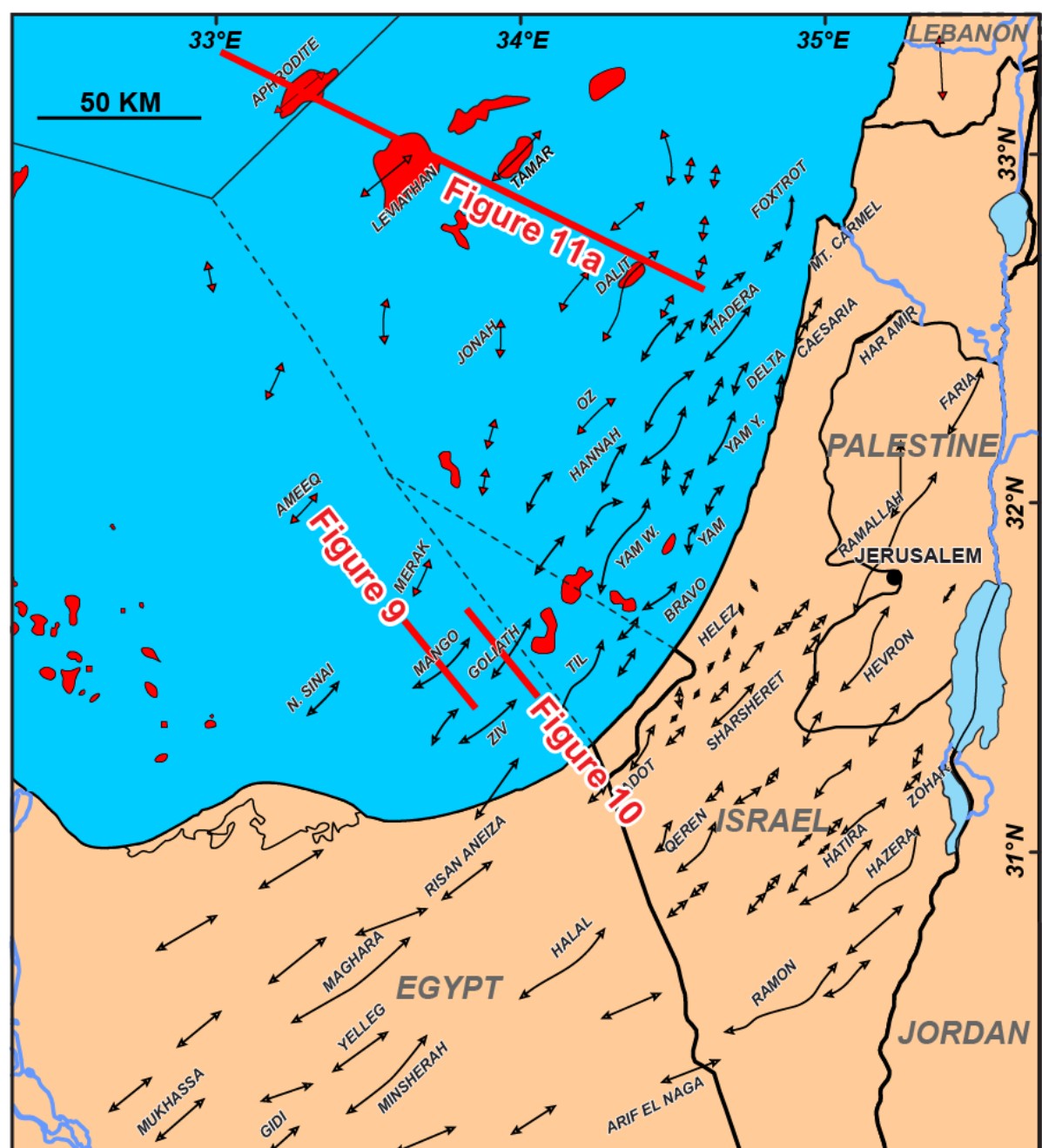

**Figure 8.** Inventory of the Syrian Arc anticlines, in the border region between Egypt and Israel, modified from Gardosh and Tannenbaum (2014) with some additions in the Egyptian offshore. The anticlinal axes in black correspond to the early Syrian Arc deformation (Late Cretaceous to Paleogene), whereas the red ones formed during the late Syrian Arc inversion (Late Cenozoic). Note the location of the Mango, Goliath and Tamar anticlines in Egypt and Israel (Figs. 9, 10 and 11), respectively.

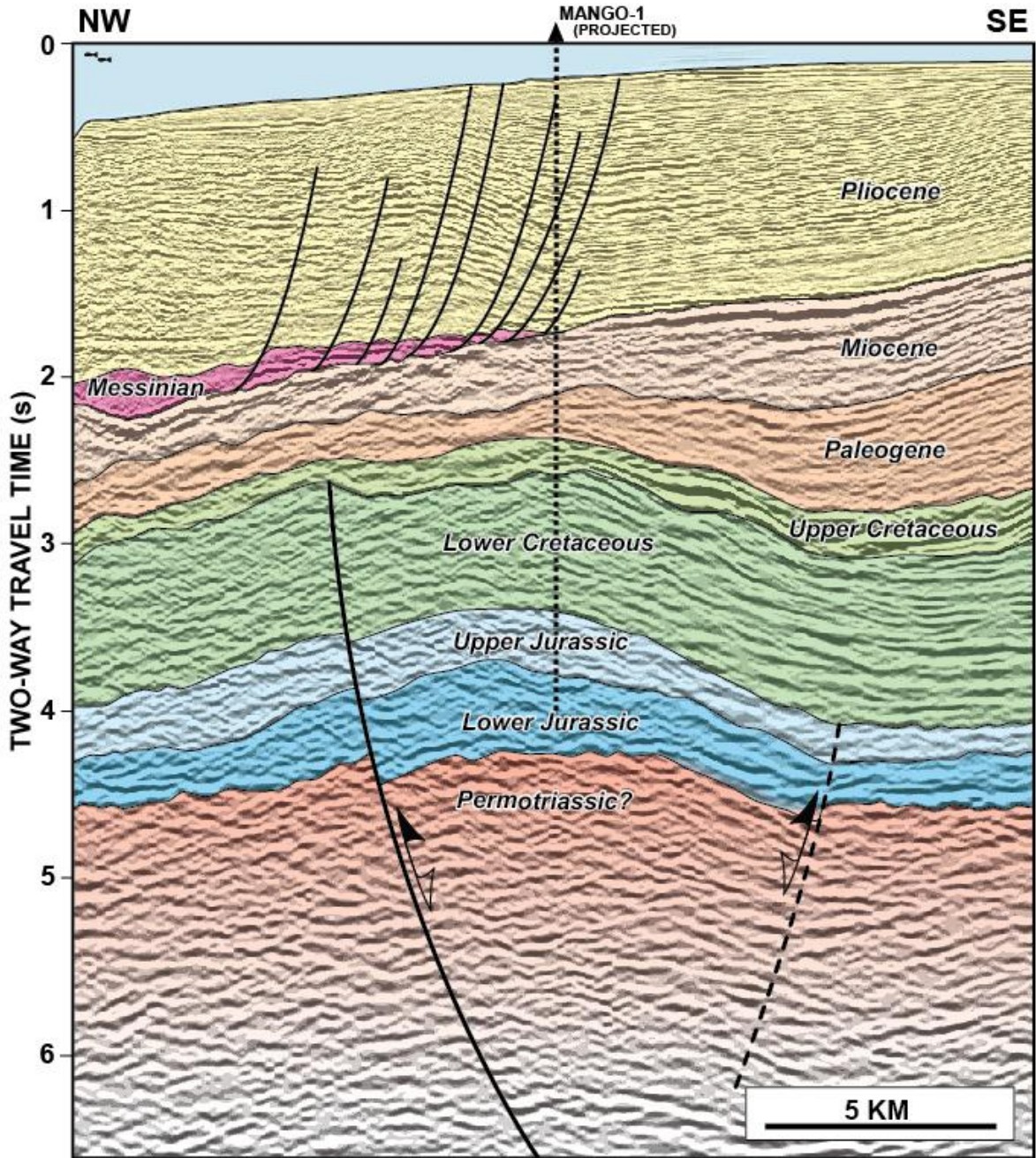

**Figure 9. 2D reflection seismic profile across the Mango oil discovery in offshore Sinai, Egypt (see location in Fig. 8). The syn-rift master fault controlled the deposition of the Lower Jurassic to Lower Cretaceous sequence. The subsequent Late Cretaceous to Paleogene Syrian Arc inversion period is responsible for the formation of the asymmetrical anticline (modified after Yousef et al., 2010).**

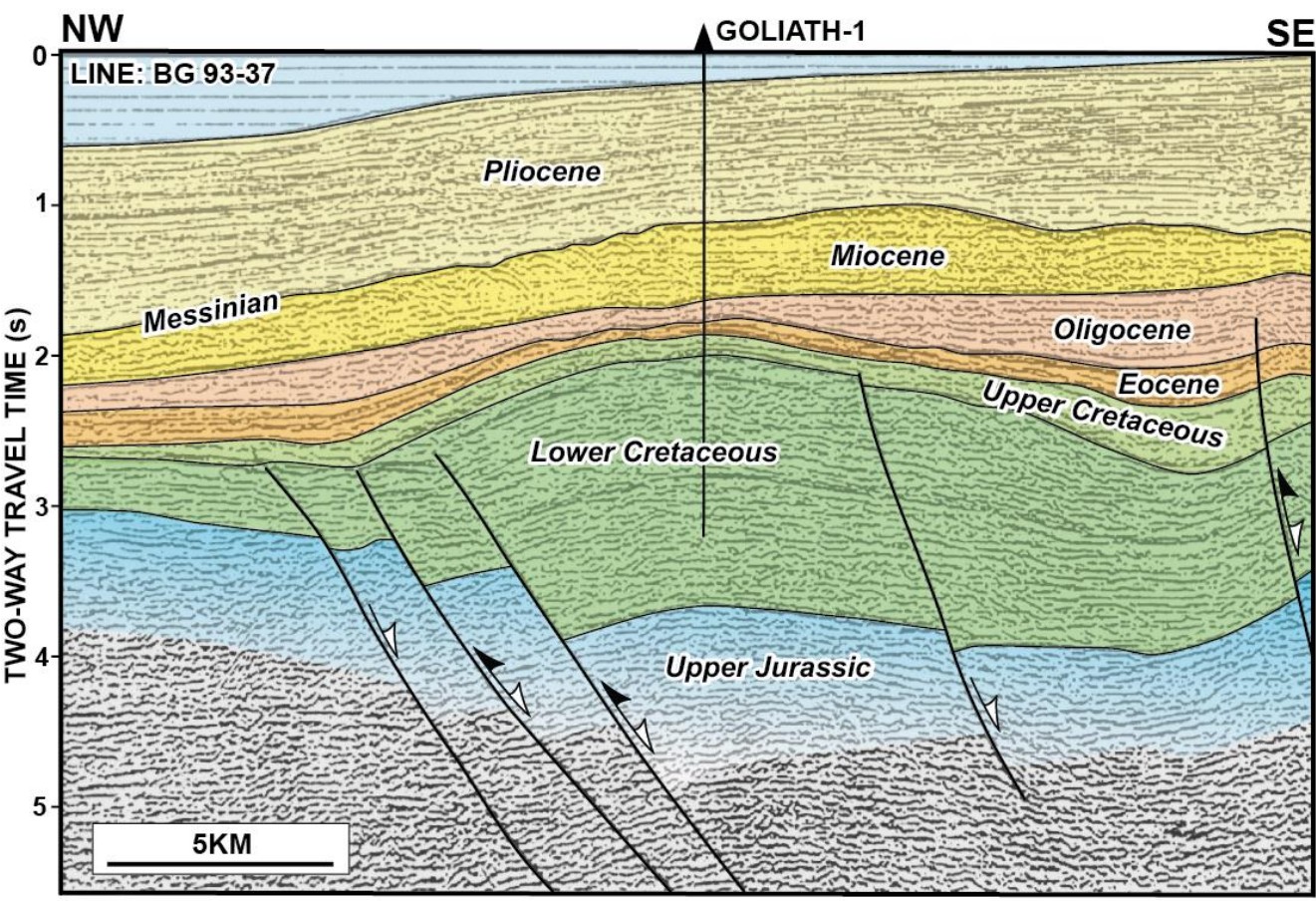

**Figure 10. 2D reflection seismic profile across the Goliath anticline in offshore Sinai, Egypt (see location in Fig. 8). The large syn-rift master fault controlled the deposition of the Lower Cretaceous sequence. The subsequent Late Cretaceous to Paleogene Syrian Arc inversion I is responsible for the formation of the asymmetrical anticline (adapted from Ayyad et al., 1998).**


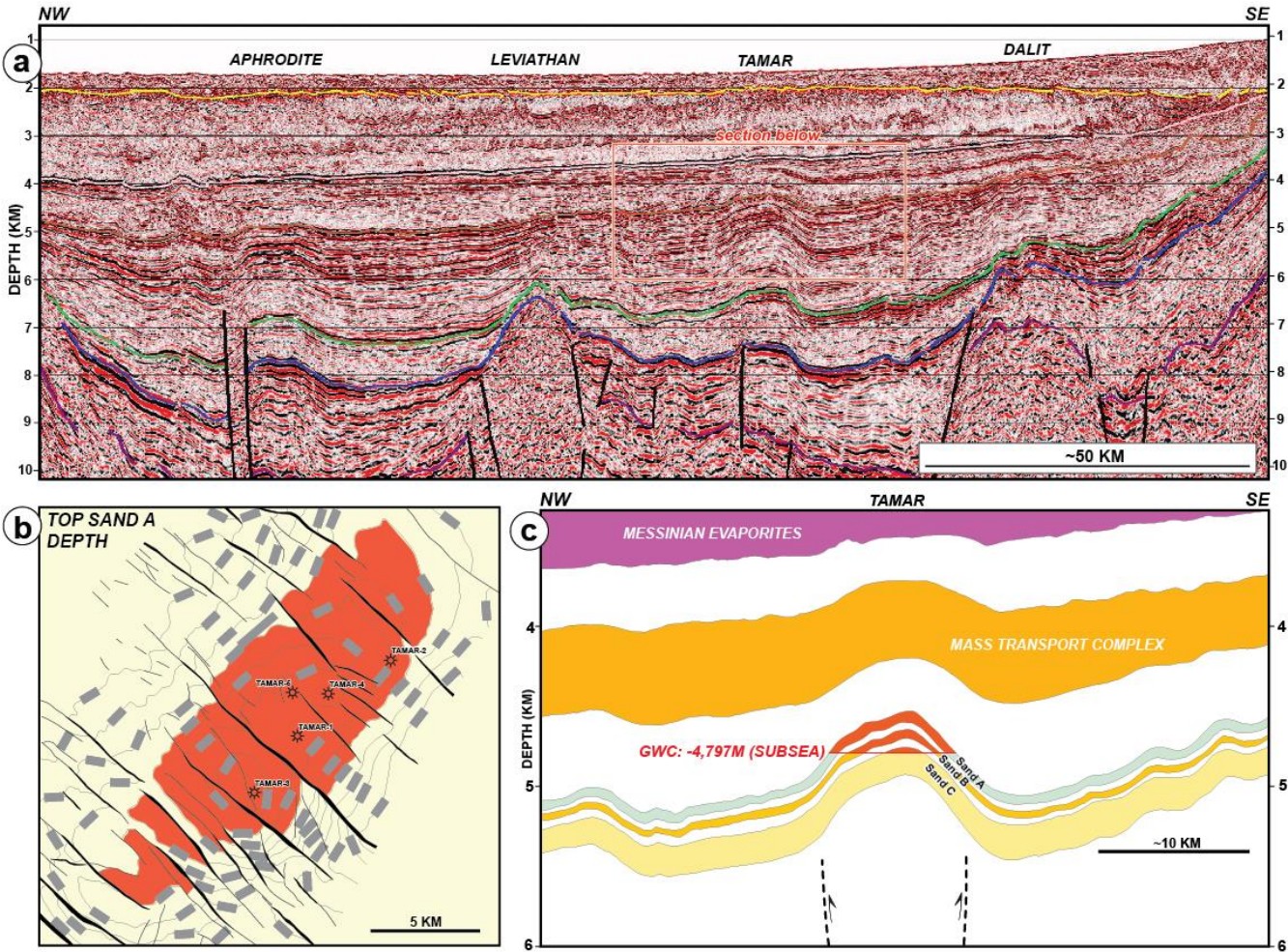

**Figure 11: Highlights of the Tamar field, offshore Israel, for location see Figure 8. a) Regional-scale pre-stack depth migrated seismic reflection section across the Levant Basin in offshore Israel and Cyprus (Roberts and Peace, 2007). Note the large vertical exaggeration. b) Structural depth map on the Miocene Sand A reservoir level (Needham et al., 2017). c) Simplified cross section across the field based on Needham et al. (2017). Note the large vertical exaggeration.**

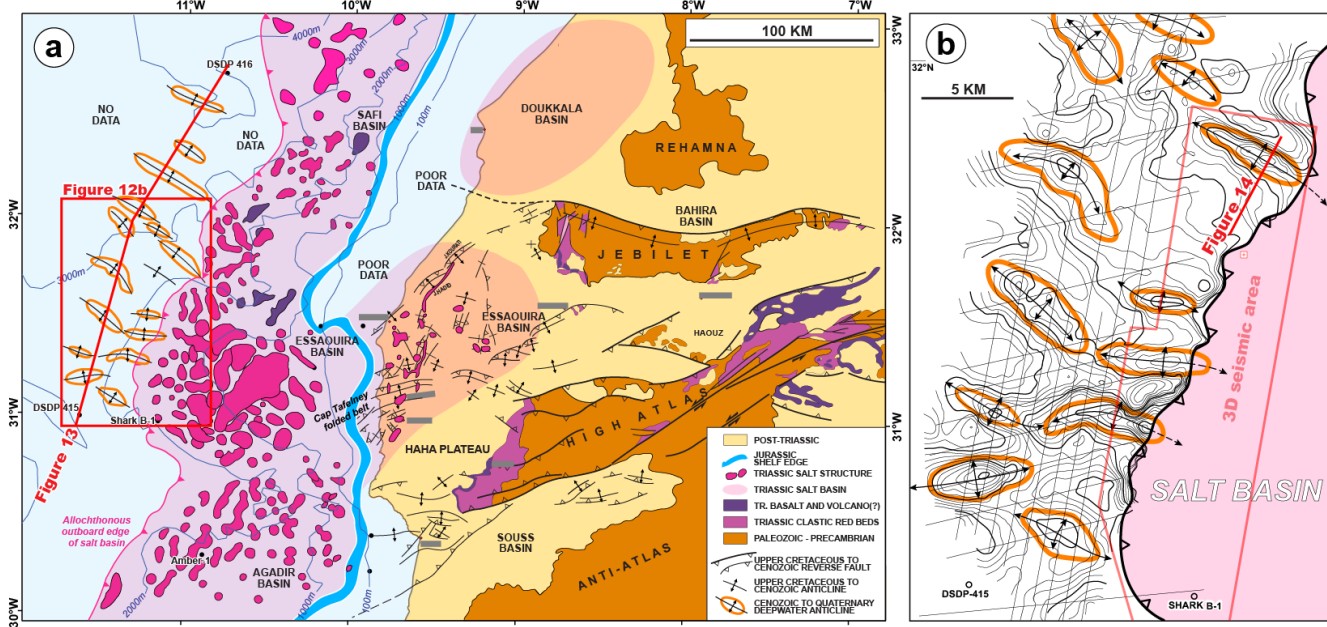

**Figure 12: a)** Simplified structural map of central Atlantic Morocco, modified from Tari et al. (2017a). **b)** Time
structure map on a mid-Jurassic mapping horizon using 2D seismic data located just outboard of the salt basin
(shown in light magenta). The prominent late Cenozoic inversion anticlines are shown by orange outlines.

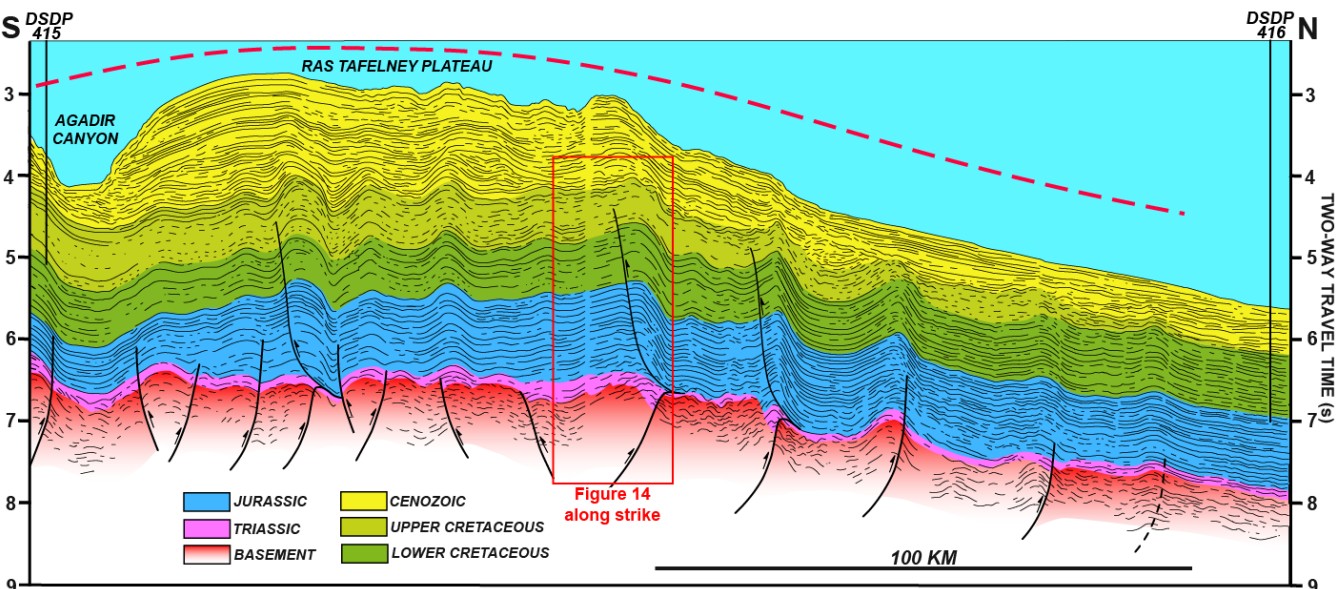

**Figure 13: Line drawing interpretation of a composite regional 2D seismic transect adapted from Tari et al. (2012).**
**Line drawing interpretation of a regional cross-section connecting the DSDP 415 and 416 wells. Note the expression**

of middle to Late Cenozoic inversion which is driven by the compressive reactivation of basement involved normal faults and linked detachments within the Mesozoic basin cover. The dashed red line shows the regional for the upwarping of the offshore continuation of the Atlas system. The approximate location of a nearby industry 3D seismic section (Fig. 14) is highlighted by a red rectangle.


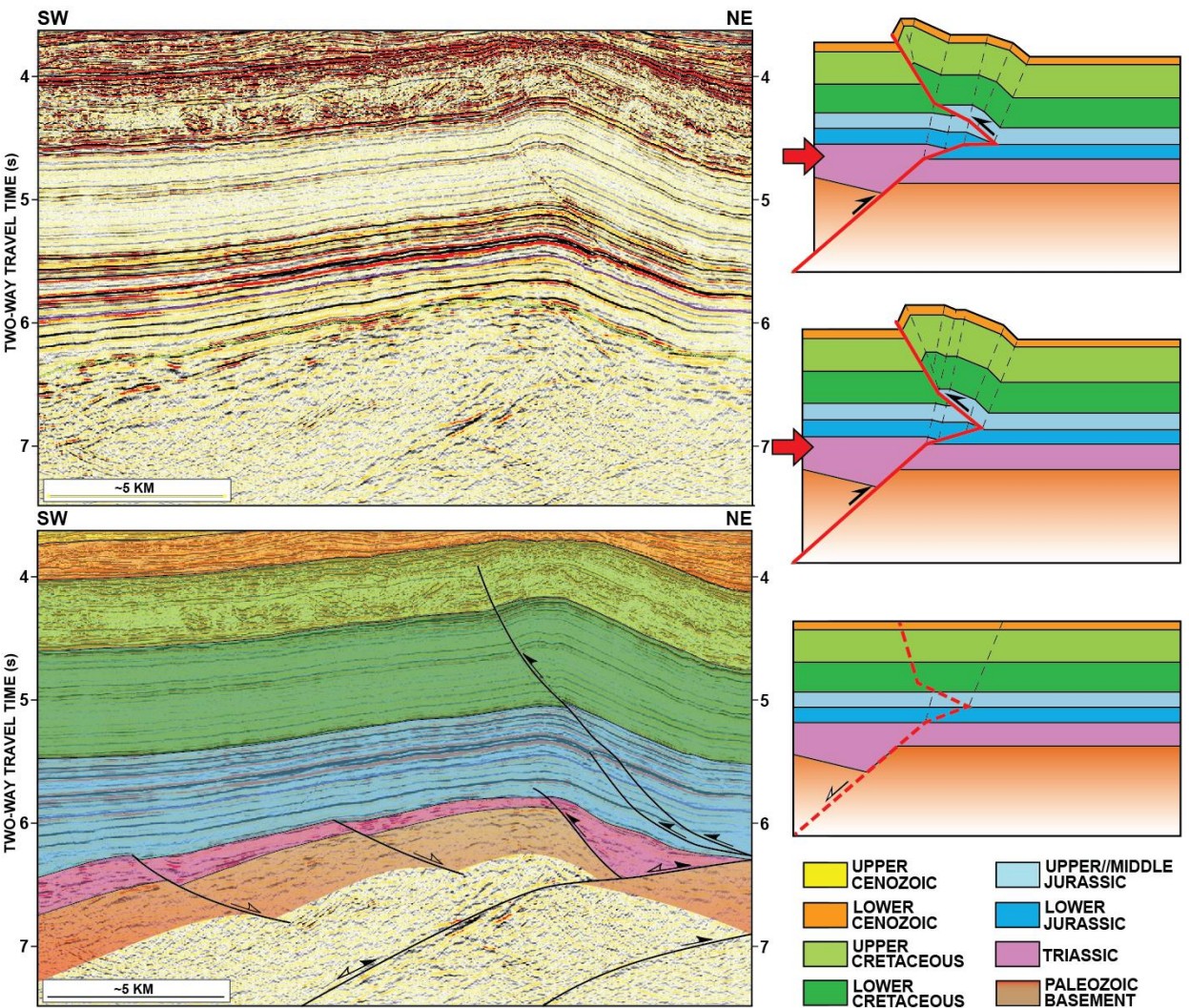

Figure 14: a) Dip directed 3D seismic reflection seismic line across one of the inversion anticlines in deepwater Morocco, for location see Figs. 12b and 13. b) The formation of a basement–involved inversion fold adapted and slightly modified from McClay et al. (2019). This structural model is the explanation for the somewhat unusual inversion geometry for some of the deepwater anticlines in the Ras Tafeleny Plateau area, offshore Morocco (Fig. 13).

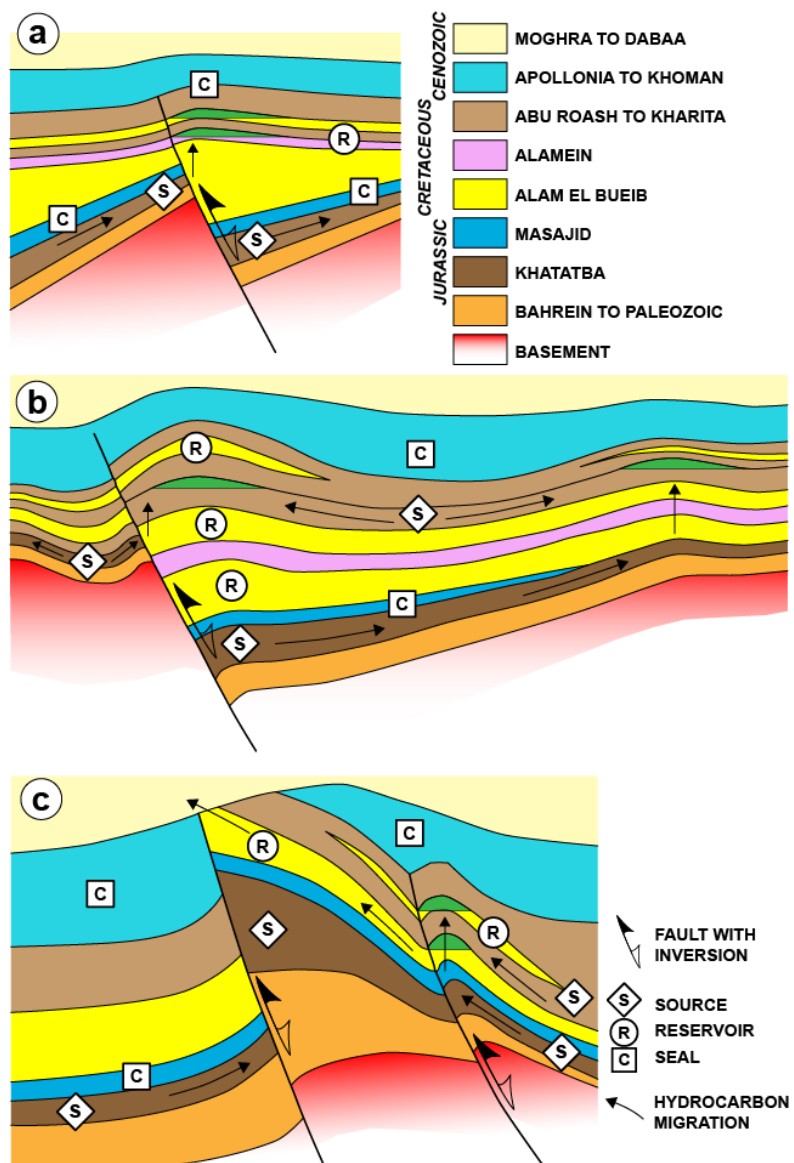

**Figure 15: Cartoons of potential charge scenarios of structures associated with inversion structures, redrawn from Bevan and Moustafa (2012). These examples from the Western Desert of Egypt illustrate well the increasing severity of inversion with the corresponding variations on the petroleum system migration patterns. The three progressive stages are shown using the oil field examples of (a) Razzak, (b) Mubarak and (c) Kattaniya. See text for a generalization of the impact of inversion tectonics on petroleum systems and exploration efforts.**

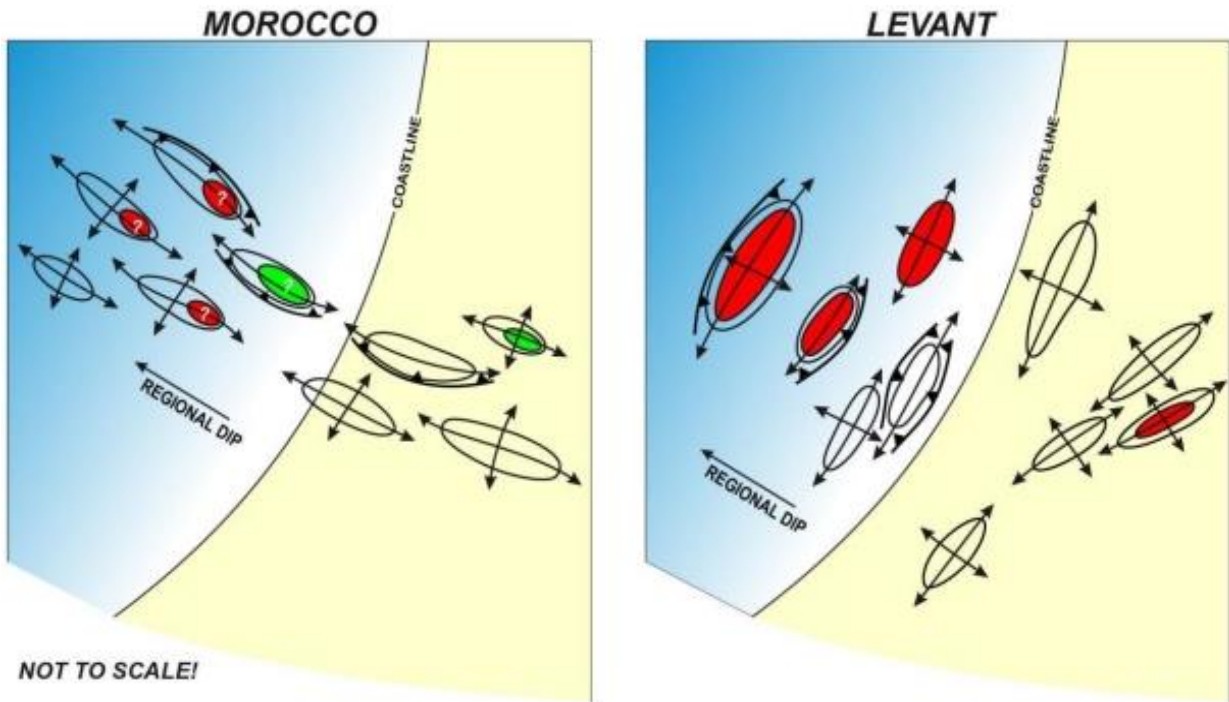


**Figure 16: The impact of the trend of inverted structures versus the regional dip, adapted from (Tari and Jabour, 2011). From a trapping point of view, if inversion structures strike perpendicular to the regional dip in a basin then it translates to an optimum situation as to the size of the four-way closures of the anticlines. In contrast, in a basin where the inversional anticlines have the same trend as the regional dip, the four-way closures on the updip end of the structures tend to be much smaller.**

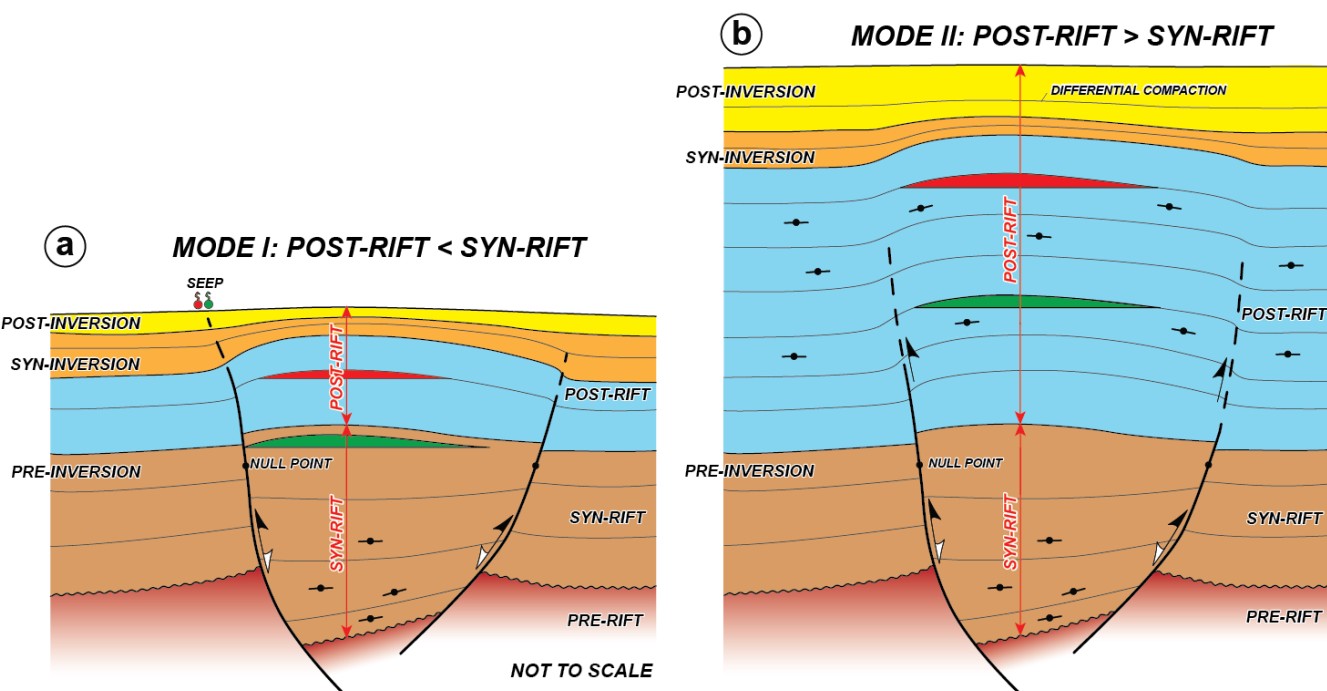


**Figure 17: Subdivision of inversion structures into two modes of inversion. A structure developed in Mode I inversion if the syn-rift succession in the pre-existing extensional basin unit is thicker than its pre- and syn-inversion parts of its post-rift cover. In contrast, a structure evolved in Mode II inversion if the opposite syn-** *versus* **post-rift (pre- and syn-inversion) sequence thickness ratio can be observed. These two modes have different impacts on the petroleum**

**system elements in any given inversion structure.**

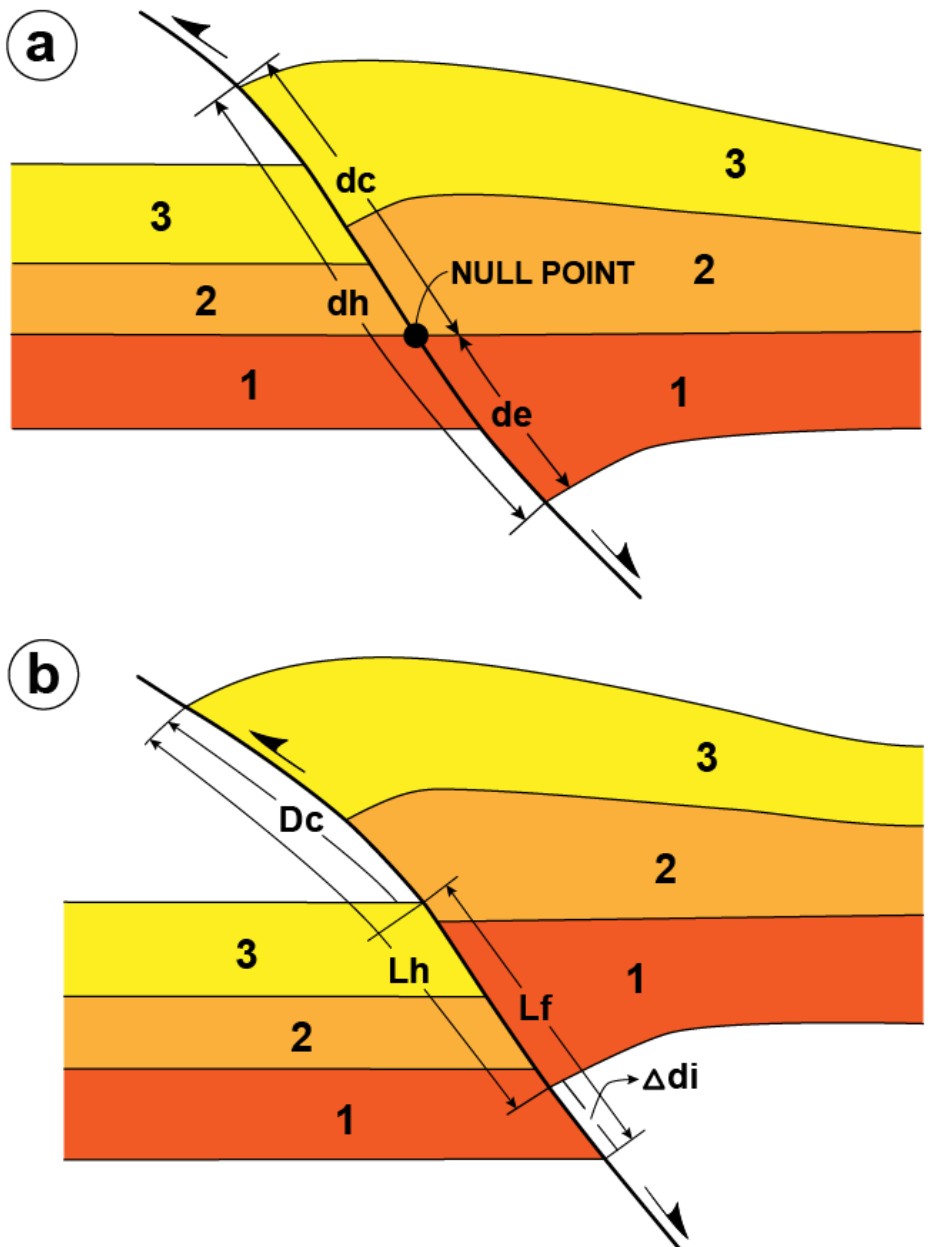

**Figure 18: Two methods to determine the inversion ratio in a quantitative manner by a) Williams *et al*. (1989) and b) Song (1997). In our view, the quantification of inversion tectonics remains a challenge as the deeper section beneath an inversion anticline is typically not well imaged seismically and/or not drilled due to the greater depth. Therefore a more practical approach is needed to describe inversion tectonics in cases where not all the required geometric elements of a structure can be measured due to subsurface data constraints.**