# Peer review of "Inversion tectonics: a brief petroleum industry perspective"

_Solid Earth, 2020_

## Referee Comment (RC1) · William Bosworth (Referee) · 18 Apr 2020

General Comments:

The differentiation between basin-scale and prospect-scale inversion structures is of fundamental importance to both hydrocarbon exploration and the interpretation of a region's tectonic and local structural histories. The manuscript does an excellent job of highlighting and explaining these differences. Documenting the relationships between the extent of inversion and the associated exploration risks and rates of success is of paramount interest to the petroleum industry. Both general themes will be of broad international and cross-disciplinary interest. The authors identify and discuss the bias in industry reporting of "inverted" versus "folded" structures and corresponding hydro-

carbon accumulations. This is an extremely interesting and important observation.

The figures are all very well done, informative and necessary. The referencing is quite extensive and up-to-date.

Specific Comments:

Section 3; circa line 115: Perhaps another contributing factor to the shift in terminology from the exploration to production domains is the nature of the geoscience and engineering teams involved. Explorationists are typically stronger in subjects such as regional geologic synthesis, tectonics, paleogeography, etc. Development and production teams focus on detailed fault patterns, reservoir connectivity, fluid flow, etc. Whether a structure originated via straight-forward contraction or inversion might not even be part of a development geologist's everyday working vocabulary, nor critical to how they develop and produce a field.

Section 6: another positive aspect of inversion features relates to intra-basinal folds that form during inversion events early in a rift basin's structural history. The authors have described inversion structures that formed after a major phase of extension (the classic geometry). But most of the basins of North Africa are actually multi-phase in origin. This has resulted in the formation of inversion-related traps that were subsequently covered by later generations of syn-rift strata. Hence, these old anticlines pre-date any migration, were covered by later top seal even if initially breached, lie above the first phase of source rock deposition, and are perfect exploration targets. The best example of this in North Africa's petroliferous basins is the "Cimmerian" event, discussed briefly by Lucic and Bosworth 2019, although they don't go into trapping details or specific examples. So for multi-phase extensional basins intervening periods of inversion can be a very good thing.

Figure 8; the "limit of Syrian Arc structures" is perhaps a bit misleading. Very nice folds and reverse faults and an associated unconformity have been documented in the southern Gulf of Suez (late Santonian main phase shortening; Bosworth et al., 1999,

Geology v. 27). Since that paper was written we have found several other even larger examples (wavelengths 100's of meters), previously attributed to other deformation mechanisms/events. But all of these are of course much smaller than the classic features on your map. Your limit is for kilometric scale structures; it makes sense that the amount of deformation gradually declines to the south and not abruptly. The smaller central and southern Gulf of Suez features are not necessarily "inversion" structures sensu stricto, but they are Syrian Arc structures in the broader sense of the term.

Minor Technical Corrections:

Line 34: "lots of work" might be replaced with "significant work" or "numerous studies". Line 77: this is a half graben not a graben in the figure, both b and c. Line 116: I don't think "overrated" is appropriate. "Overstated" would be better. Line 135: change "from the W" to from the west. Line 153: "oriented perpendicular". Line 157: change "enough the properly" to "enough to properly". Line 238: "which was deposited". Line 244: change "there are lots of publications are devoted" to "numerous publications are devoted". Line 301: "their view might also be somewhat". Line 302-303: "there is plenty of evidence".

---

## Referee Comment (RC2) · Mark Rowan (Referee) · 20 Apr 2020

General Comments

This paper is a well-written review of those aspects of inversion tectonics most relevant to the petroleum industry. As such, it is not intended as an exhaustive review, but instead highlights certain aspects with some specific examples. The paper succeeds in this effort, providing a clear summary and important insights. The examples used are not the highest-quality seismic images of inversion structures, but they were chosen to support the message offered in this paper and are more than adequate. A welcome addition is the authors' emphasis on just why it is valuable for industry to properly identify prospects and fields that have an inversion origin, and why a more quantitative

and systematic approach is needed in the future.

Having said that, there are some relatively minor issues that, if addressed in the paper, would only add to the quality and applicability. These are itemized in the specific comments below.

Specific Comments

1. There is an almost exclusive focus on subsurface data in identifying inversion structures (although regional understanding of the basin evolution is also emphasized). Even in the petroleum industry, surface observations can provide important clues. For example, significant thickness and facies changes between the footwalls and hanging walls of thrusts can provide clear evidence of preexisting extensional faults. Classic examples include the southern Pyrenees of Spain and the Eastern Cordillera of Colombia (which is a major petroleum province). In the latter case, two papers had very different interpretations of essentially the same cross section: whereas Dengo and Covey (1993) interpreted most of the structures as thin-skinned deformation, Cooper et al. (1995) used, among other criteria, dramatic facies and thickness changes within synrift fill to demonstrate the preeminence of inversion tectonics. It would be good to add some comments about this additional tool in recognizing inversion.

2. It would also be beneficial, especially because Bally's figure includes it, to cover briefly some of the effects of salt during inversion. One is the decoupling of deformation above and below (e.g., Letouzey et al., 1995), which can complicate using observed large-scale geometries to recognize inversion or lead to secondary, intrabasinal traps (Stewart, 2014). Another is the varying impact of prerift, synrift, or postrift salt on the petroleum system, given its tendency to act as a barrier to upward migration.

3. Lines 80-81. Is this really a common problem? Forced folds are monoclines, with one side below regional, whereas inversion structures are anticlines that are above regional.

4. Lines 112-113. It's not clear just what data support this statement that "inversion" is used quite loosely in exploration. You have used the data mining to show that traps are rarely termed inversion structures, but this is just a statement. It's the same on Line 116 when you state that inversion tectonics appears to be overrated in exploration. Please clarify.

5. Section 4, first paragraph. Can you explain or speculate on what is generating the two different wavelengths of inversion structures? Distinguishing between regional and local inversion was highlighted in the introduction, so what is causing the differences in this case? Also, it's not clear how the map view pattern of the shorter-wavelength structures suggests an eastward propagation. Only timing data can tell you this. And finally, what are the orientations of the stresses you cite in the last sentence?

6. Lines 168-169 and Fig 6. It's very difficult to see the asymmetry with so much vertical exaggeration – I suggest redrafting with much less distortion. Also, it seems that the best indication that this is an inversion structure is not the asymmetry, but the thickness pattern in the Badenian and older strata. This should also be cited in the explanation around Line 175 and in the figure caption.

7. Lines 197-198. Salt-detached structures can still be part of inversion structures. So is there no basement inversion in this (Syrian) part of the Syrian Arc? Or could it be decoupled inversion due to the influence of salt? Also relevant to the caption for Fig 8.

8. Line 201. It says that the Syrian Arc includes part of the Western Desert, but Line 187 says from Sinai to the Palmyrides. Needs to be consistent. Also, it would be helpful to mention the time of rifting in the paragraph starting on Line 196. There's a mention on Line 207, but it's not clear whether this applies to the whole area or not.

9. Lines 212-215. The Syrian Arc I structures are asymmetric, but what is used to identify inversion on the more symmetric Syrian Arc II structures?

10. Lines 219-220 and Fig. 10. The text says that the NW limb is steeper, but the

figure shows what looks like a shorter, steeper SE limb (especially if Pliocene tilting is removed). Some might say that it's symmetric, so what criteria are being used to say this is in inversion structure? Is it the regional setting? If so, you have also cited the presence of folds detached on Triassic salt, so how can you distinguish between them?

11. Section 5.2. The geometries in Fig 11a are inconsistent in terms of clear inversion. This may be beyond the scope of this paper, but: Leviathan is the most obvious inversion structure, with a rift hanging wall inverted above regional; the prominent basement highs on either side of Tamar are bounded by uninverted rift basins and are themselves long-lived highs; and Tamar itself is enigmatic, with no clear underlying rift-bounding fault. So it really comes down to what you say in the last sentence, namely that an inversion interpretation is based by regional timing. You might add the similar timing of formation to known inversion structures.

12. Lines 262-265. This seems a little contradictory and thus maybe should be reworded. First it says that the source rocks in the hanging wall are critical. But then it says that hydrocarbons from here might migrate away from the trap, and that it is the footwall source rocks that are more important (as shown in Fig 12a and b). There is a similar problem in the conclusions (Line 337).

13. Lines 314-315. Figs. 7 and 11 are cited as examples where the source rocks are in the postrift fill, but don't actually show anything about the stratigraphic level of the source rocks. Either indicate these levels on the figures or remove this from the text.

14. Please indicate the locations of the Budafa and Lovászi fields on Fig 3.

15. There are a few places (e.g., Fig 13) with redundancy between the text and the figure caption. This should ideally be removed.

Technical Corrections

Marked-up manuscript with minor corrections sent to author separately.

---

## Referee Comment (RC3) · Michael Gardosh (Referee) · 1 May 2020

General comments:

1. Unifying theories and generalized concepts form an essential part of our 'tool-box' as earth scientists and explorationists. Their predictive power is extremely important to our ability to reduce risks and increase the chances of success in looking for oil and gas reservoirs. The authors are doing a good job in highlighting the need for such concepts in the context of Inversion and the investigation of Inversion Structure with a petroleum system perspective in mind. Some of the principle relations between structural evolution and hydrocarbon accumulations in selected case studies are well presented and summarized in this manuscript, thus forming an excellent basis for the

continuation of research in this highly important subject for the petroleum industry.

2. Although the goal of the manuscript is to provide a high-level view on the subject matter it is focused on the later, contractional phase of the Inversion story. The review can benefit from the addition of some more details on the potential source rocks and migration patterns of hydrocarbons associated with the case studies discussed, where available. Although these are typically more difficult to describe, their inclusion can clarify to the reader the relations between the extensional and contractional phases as shown in figure 12 or their lack of, in the various examples presented.

3. The authors selected the Syrian Arc structures of the Levant region as a principle case study for their review. This selection is understood, as these structures are important from the industry's point of view, hosting some of the largest accumulations of gas found in the last 12 years worldwide. However, it should be noted that while Syrian Arc structures were extensively investigated by many workers in Egypt, Israel, Lebanon and Syria the understanding of their overall tectonic evolution is still incomplete. The age of the extensional phase in various individual structures is ranging from late Paleozoic to Mid Jurassic and in some structures there seems to be negligible or no extension (e.g. southern Israel, Palmyrides), whereas the contraction is ranging from Late Cretaceous to Late Neogene; in both cases roughly 100 million years for each. In view of this complex geologic history it appears that lessons learned from a specific case study can be applied only to other structures with similar geologic characteristics and not to the entire fold system.

Specific Comments:

1. Line 123- Should be . . .Hungary: Lovaszi and Budafa oil and Gas Fields.

2. Line 190- the age of Syrian Arc deformation cited from Walley (1998) is incorrect. Walley refers to a first phase of contraction during the Coniacian-Santonian, while its second phase is of Late Eocene to Late Oligocene. It should probably be added here also that later studies, particularly offshore Israel show the second phase to extend

until at least the middle Miocene as described by Needham at al., (2017) and in chp. 5.2.

3. Line 214- ….are found predominantly in the offshore part……as there are other Syrian Arc structures inland (Israel and Syria) that were active during the late contraction phase.

4. Line 224- the recoverable reserves in the Tamar Field noted by Needham at al (2017) are ∼11 TCF (for late 2016). In early 2019 the Israeli Ministry of Energy estimated the reserves as 8.5 TCF.

5. Lines 335-336 - The authors state, "closures typically cluster above the extensional depocenters which tend to contain source rocks providing petroleum charge…...". This conclusion is not well supported by the case studies presented.

6. Lines 351-354 highlight the "important of source rocks not being constrained to the syn-rift basins" in contrast to the previous statement (Lines 335-336). The ms will benefit from a more comprehensive discussion on the position of source rocks and their relation to the Inversion story in various cases. Two end-members may be described one with source rocks within the inverted system and one outside of it.

7. Chp. 8- This chapter is not a typical Conclusion chapter but rather an extension of the discussion. It is advised to re-write it summarizing the previously discussed issues.

Comments on Figures:

Fig. 3- The right-hand side of the map does not include any relevant information, if possible crop it out. Mark the location of Sava Folds and Budafa+Lovaszi Fields/Structures. Mark the location of seismic lines and sections in following relevant figures.

Fig 4- Add well names or field names in seismic section.

Fig 5- Indicate location of seismic line in map.

Fig 6- Indicate location of section in map.

Fig 7- Label thickening/thinning geometries in seismic lines or describe in caption.

Fig 8- Correct misspelling in caption- Israek.

Fig 14- Add description to abbreviations and explain in the caption or in text the difference between the two methods of calculation.

A version of the text with additional minor corrections was sent in PDF file to G. Tari.

---

## Referee Comment (RC4) · Gábor Bada (Referee) · 14 May 2020

General comments

The paper of Tari et al. presents a brief yet comprehensive overview on the exploration aspects of inversion tectonics. The authors had to follow a tricky path as to the level of details which is inevitably a balancing act for review type papers. The topic and the wide range of its implications for petroleum exploration merits the size of a textbook. Yet the paper successfully navigates through some of the key features of basin inversion such as trap formation, charging, maturation, etc., and their temporal aspects.

Continuous improvement in seismic imaging has helped significantly the reconstruction of 3D subsurface anatomy, both tectonics and stratigraphy. This presents a good op-

portunity to revisit the concept of basin inversion from a phenomenological, kinematic and dynamic point of view. Indeed, the use of the term "inversion" has been quite slack ever since its introduction some 40 years ago. The authors recognize this and make a good attempt to explore reasons and consequences. Ultimately, the questions comes down to this: i) is the original definition of basin inversion good enough for practical use by the industry and ii) how can we make a shift from (semi-)qualitative analysis to more quantitative interpretation and ultimately more successful prospecting? The paper provides essential insights to get closer to answering these questions.

On a side note, publishing A. Bally's unpublished work (Fig. 1) is a timely gesture after the pass away of this visionary geoscientist last year.

In conclusion, the paper by Tari et al. is a well-presented scientific contribution with practical significance and inferences.

Specific comments

The paper needs a somewhat better definition of scope and rationale. What is the focus: more descriptive or process oriented or covers both?

The authors' definition for inversion (lines 73-74) is inevitably a bit loose which is inescapable. What is the scale of anticlines? Would fault-bend anticlines qualify whereby main deformation is taken up my reverse faulting rather than 'pure' folding? Where is the boundary between complete basin inversion and onset of (over)thrusting and nappe formation in former extensional settings (passive margins becoming active margins to the extreme). Little chance to get to a generally accepted definition though.

The authors present the seemingly surprising statistics of underreported inversion cases for trap forming mechanism using the IHS Markit database. This goes back to the practical value of its definition. Compression is generally easy to recognise but applying the definition of inversion, whatever that is, is another matter. Support with data the statement that "inversion tectonics appears to be somewhat overrated in exploration" (lines 115-117).

The Pannonian case studies are well presented and discussed. Large-scale uplift of Transdanubia deserves some more details. Is it due to lithospheric/crustal buckling as a result of horizontal compression, or perhaps isostatic adjustment due to lateral variation of preceding extension? In addition, as shown in several cases in a hot yet thermally disturbed lithosphere a few hundreds meters of uplift has a significant footprint on maturation. What is the impact of 800 to 1,000m estimated uplift at the Lovászi field? How do inverted extensional faults and newly propagating reverse faults behave in terms of fluid migration and subsequently what is their seal capacity? These aspects may merit a few paragraphs of additional discussion. Figure 7a needs visual improvement. Highlight with colour the main stratigraphic units so that "... the thickening/thinning geometries within the Upper Pliocene (Pannonian) strata in the apex of the anticline show the switch from extension to compression" (lines 175-177) hits the eye and hence becomes more apparent for the reader. The EastMed case studies are also educative – well written with sufficient details. Figure 9 is hard to read, increase resolution (presumably, printed version will be clearer) and find more distinct symbols for depicting the two inversion phases.

Sections 6 and 7 provide an excellent summary on the implications of inversion on petroleum systems and exploration. A couple of specific discussion points to consider further:

- Negative impact of inversion tectonics: even significant uplift, if reservoirs sealed properly, or self-sealed, may have positive impact such as enhancing reservoir energy (overpressure) and dewatering (tight gas in the Rockies), gas segregation in biogenic setting (see examples in PanBas), etc.

- Extend discussion on the impact of thickness difference of syn-rift vs post-rift sediments.

- Inversion often results in anticlines high above blind reverse faults within the 'basement' – clarify further how charge occurs in such unfaulted traps.

- Elaborate further the role of the trend of inverted structures vs regional dip as per Tari and Jabour (2011). The current paper presents only a brief summary of this relevant topic without explaining the causes.

---

## Author Comment (AC1) · 5 Jun 2020

Response to the comments made by Bill Bosworth (Referee) We really appreciate the positive and constructive comments made. We will incorporate all the required changes into the final version. Here are the specific responses to the comments. Specific comments: Section 3; circa line 115: The explanation given by the referee is much better formulated than what we offered and, admittedly, we will use it for the final version, as is. Section 6: We agree and will reference not only the paper by Lucic and Bosworth, but also the paper by Bosworth and Tari in this same issue of Solid Earth. Fig. 8. we agree that intuitively, the southern limit of the Syrian Arc inversion belt is a not a well-defined line but rather transitional zone depending on the scale of observation. This important point will be also added to the text and the corresponding reference as

well. Will be reworded accordingly. Minor technical corrections: Instead of listing them here, let us just say thanks for spotting these mistakes. All of them will be corrected accordingly. Finally, we also appreciate the informal insights offered by the referee on several occasion during the last few years into the inversion process in the broader North Africa region.

---

## Author Comment (AC2) · 5 Jun 2020

Response to the comments made by Mark Rowan (Referee) We very much appreciate the constructive and helpful comments and also the extra effort to go through the draft document in details. All the comments were taken on board, one way or the other. Here are the specific responses to the comments made. 1) We agree with the statement that surface observations cannot be ignored when it comes to identifying inversion. However, we believe that in the common industry practice surface geology constraints are considered as a luxury and that is why we focus our overview on subsurface examples. Many of the basins we keep working on are either located offshore and/or have a post-inversion sedimentary blanket which rules out field work as a meaningful additional source of observations. Yet, we will acknowledge the existence of case studies

where surface work can make a critical difference, like in the examples mentioned by the referee. 2) We deliberately tried to stay away from the topic of salt tectonics versus inversion as it is a very complex subject on its own with lots of relevant work done by many folks in the last few decades. We are just as aware the importance of this field as the referee, so we will make a short detour in the text to highlight this class of inverted structures. 3) In our industry experience these two different class of structures are mixed up easily. But the point made by the referee is a very good one which we will shamefully take as is, with a reference to him. 4) Again, working in the petroleum industry, we quite often see statements about inversion made in a very loose way. It is not our goal to quote many examples from published literature where structures are referred to as being "inverted", even if the structures are not even remotely fulfilling the original definition of structural inversion. In our experience, inversion is generally used as a positive "selling point" when it comes to prospect evaluation. As another referee (Bill Bosworth) pointed out, perhaps "overrated" should be replaced "overstated". Our data implies not necessarily the overstated perception of the positive effects of structural inversion by explorationists but rather the disinterest by petroleum engineers who inherit the project at a later stage. 5) These questions posed by the referee will be answered in the final version. 6) An easy fix and we will redesign the figure with less vertical exaggeration. We just redrafted the original as is, but indeed it should be stretched horizontally. 7) This is not so easy to answer based on the published literature. There are some cases in the Palmyrides where the basement is shown to be involved, but in our view this folded belt is largely detached on the Triassic salt and, therefore, should not be regarded as the simple prolongation of the "Syrian Arc" inversional belt in the Levant region. 8) Will be reworded, given the issue mentioned above. 9) This meant to be one of the messages in this paper, i.e. the distinction between the more classical looking Syrian Arc I structures and the Syrian Arc II structures in the offshore Levant where the thick post-rift sedimentary cover results in more subtle asymmetry of the structures. The giveaway for the inversional origin of these anticlines is the presence of coinciding underlying Triassic(?)-Jurassic rift basins. 10) The pub-

lished seismic line across Mango is indeed not very convincing on its own. Therefore we will replace it with another published seismic line across the nearby Goliath inversion structure which is a much clearer example of the process. 11) We agree with these statements. Here we are constrained by what has been published as to regional seismic examples. On the published line we use, Leviathan is clearly a much better example. As to the interpretation of Tamar, we had access to much more seismic data in this area to be certain that there is, indeed, a coincidence between an underlying rift basin, with its master fault, and the Tamar structure high above it. We will highlight the basin-scale analogy using the inboard examples documented by Gardosh, for example. 12) Correct, will be reworded to remove the contradiction. 13) Correct, will be shown on the figures. 14) Correct, will be done. 15) Well, it is a matter of style as many people these days will only glance at the figures and expect to have a fairly complete explanation in the captions... But we will shorten the captions to minimize the overlap with the text. Again, we acknowledge here the extra effort by the referee correcting many other minor items in the text/figures and sending the annotated draft directly to us. We consider this effort by the referee quite exceptional these days and we are very grateful for it!

---

## Author Comment (AC3) · 5 Jun 2020

Response to the comments made by Michael Gardosh (Referee) We really appreciate the positive and constructive comments made. We will incorporate all the comments into the final version. Here are the specific responses to the comments. General comments: 1. We agree, this is a highly important subject for the petroleum industry. 2. We will add more on the topic of source rocks and charge as to the case studies. 3. We agree, extrapolation of the findings from one well-constrained inversion structure to another, less constrained one should be done carefully and indeed, it assumes a largely similar, if not identical, geological history. Specific comments: 1. Will be corrected. 2. Will be reworded accordingly. 3. Will be added. 4. The 8.5 TCF reserve number will be added quoting the referee. 5. Correct, we will have to make this clear, that "typical"

[Figure]

refs to the classical case (ratio of syn-rift/post-rift fill more than 1) whereas our case studies are deliberately showing the underappreciated case (ratio of syn-rift/post-rift fill less than 1). 6. Correct, the end member cases will be described as such. To us these are a) source rocks in the extensional basin fill and b) regional source rocks in the post-rift but pre-inversion basin fill. 7. Will be redesigned accordingly. Comments on figures: Fig. 3. Correct and it will redesigned accordingly. Fig. 4. Will be done. Fig. 5. Will be done. Fig. 6. Will be done. Fig. 7. These geometries will be noted on the figures with additional text. Fig. 8. Will be corrected. Fig. 14. It is somewhat difficult as the full explanation of the calculations would require quite a bit of additional text. We will try to highlight the overall difference between these two approaches with reference to the original publications. We will also highlight the fact that in everyday practice these are difficult calculations as many of the parameters may not be determined precisely due to poor data. Therefore, we are planning to propose a less quantitative but perhaps more practical method to categorize the degree of inversion. For example, the ratio between the elevation of the highest point of the inverted sequence above the corresponding regional stratigraphic surface and the horizontal extent of the initial extensional basin in a dip direction, on top of the extensional sequence. These parameters are relatively easy to measure and do not require the full imaging of the master fault and the various offsets along it. The geometric ratio could be used as a semi-quantitative descriptor of the degree of inversion, i.e. the larger numbers would correspond to more "advanced" cases of inversion. Finally, we also appreciate the extra effort by the referee correcting some other mistakes we made in the original draft text/figures and forwarding the annotated version to us. We are very grateful for the additional work which went into that part of the reviewing process as well!

---

## Author Comment (AC4) · 5 Jun 2020

Response to the comments made by Gábor Bada (Referee) We really appreciate the positive and constructive comments made. We will make all the required changes to produce a final version. Here are the specific responses to the comments. Specific comments in italics and answers: "The paper needs a somewhat better definition of scope and rationale. What is the focus: more descriptive or process oriented or covers both?" We will try to spell this out better in the introduction. It is a more like a descriptive and practical approach what we aimed for in this decidedly brief overview. "The authors' definition for inversion (lines 73-74) is inevitably a bit loose which is inescapable. What is the scale of anticlines? Would fault-bend anticlines qualify whereby main deformation is taken up my reverse faulting rather than 'pure' folding? Where is

the boundary between complete basin inversion and onset of (over)thrusting and nappe formation in former extensional settings (passive margins becoming active margins to the extreme). Little chance to get to a generally accepted definition though." We want to stick to the original definition, so a fault-bend fault anticline cannot be regarded as an inverted feature as typically the thrust fault plane does not have an earlier extensional movement on it. The second part is a good question, i.e. where to draw the line between regional and "complete" inversion and development of a folded belt. This is beyond our discussion as it involves other issues as well, such as reverse faults versus nappe contacts, etc. We will not be able to address these in this paper. "The authors present the seemingly surprising statistics of underreported inversion cases for trap forming mechanism using the IHS Markit database. This goes back to the practical value of its definition. Compression is generally easy to recognise but applying the definition of inversion, whatever that is, is another matter. Support with data the statement that "inversion tectonics appears to be somewhat overrated in exploration" (lines 115-117)." A similar point was made by another referee. We cannot quantify this statement with the IHSMarkit data base as it would require pre-drill versus post-drill analysis. In our experience, In our experience, inversion is generally used as a positive "selling point" in the prospect evaluation process. Our IHSMarkit data does not capture assumed overall positive effects of structural inversion by explorationists, but rather the disinterest by development geologists and petroleum engineers who inherit the subsequent appraisal and development project. "The Pannonian case studies are well presented and discussed. Large-scale uplift of Transdanubia deserves some more details. Is it due to lithospheric/crustal buckling as a result of horizontal compression, or perhaps isostatic adjustment due to lateral variation of preceding extension? In addition, as shown in several cases in a hot yet thermally disturbed lithosphere a few hundreds meters of uplift has a significant footprint on maturation. What is the impact of 800 to 1,000m estimated uplift at the Lovászi field? How do inverted extensional faults and newly propagating reverse faults behave in terms of fluid migration and subsequently what is their seal capacity? These aspects may merit a few paragraphs of

additional discussion. Figure 7a needs visual improvement. Highlight with colour the main stratigraphic units so that "... the thickening/thinning geometries within the Upper Pliocene (Pannonian) strata in the apex of the anticline show the switch from extension to compression" (lines 175-177) hits the eye and hence becomes more apparent for the reader." Good points, we will extend the discussion on this. But, admittedly, we do not have quantitative results as it would require proper basin modelling studies on the Lovaszi Field... We are not aware of such a study to date. We believe that this situation illustrates our general point, i.e. since it is an existing field, development experts probably did not see the value of trying to address the issues revolving around the inversion-caused origin of the accumulation. "The East Med case studies are also educative – well written with sufficient details. Figure 9 is hard to read, increase resolution (presumably, printed version will be clearer) and find more distinct symbols for depicting the two inversion phases." "Sections 6 and 7 provide an excellent summary on the implications of inversion on petroleum systems and exploration. A couple of specific discussion points to consider further: Negative impact of inversion tectonics: even significant uplift, if reservoirs sealed properly, or self-sealed, may have positive impact such as enhancing reservoir energy (overpressure) and dewatering (tight gas in the Rockies), gas segregation in biogenic setting (see examples in PanBas), etc. Good points, we will extend the discussion on this. Extend discussion on the impact of thickness difference of syn-rift vs post-rift sediments. Good point, we believe that this is an original observation in this paper, so we will extend the discussion on this. Inversion often results in anticlines high above blind reverse faults within the 'basement' – clarify further how charge occurs in such unfaulted traps. Good point, we will extend the discussion on this, i.e. vertical migration, sub-seismic faulting, etc. Elaborate further the role of the trend of inverted structures vs regional dip as per Tari and Jabour (2011). The current paper presents only a brief summary of this relevant topic without explaining the causes." Indeed, we will extend this part of the paper, perhaps even adding some illustrations from Morocco.